# Can Information Flows Suggest Targets for Interventions in Neural Circuits?

**Praveen Venkatesh**[1]*, **Sanghamitra Dutta**[2]*†, **Neil Mehta**[3]† **and Pulkit Grover**[4]

[1]Allen Institute, [1]University of Washington, Seattle; [2]JP Morgan Chase AI Research;
[1–4]Department of Electrical and Computer Engineering, [4]Neuroscience Institute,
Carnegie Mellon University

[1]praveen.venkatesh@alleninstitute.org, [2]sanghamitra2612@gmail.com,
[3]neilashm@andrew.cmu.edu, [4]pulkit@cmu.edu

## Abstract

Motivated by neuroscientific and clinical applications, we empirically examine whether *observational* measures of information flow can suggest *interventions*. We do so by performing experiments on artificial neural networks in the context of fairness in machine learning, where the goal is to induce fairness in the system through interventions. Using our recently developed $M$-*information flow* framework, we measure the flow of information about the true label (responsible for accuracy, and hence desirable), and separately, the flow of information about a protected attribute (responsible for bias, and hence undesirable) on the edges of a trained neural network. We then compare the flow magnitudes against the effect of intervening on those edges by pruning. We show that pruning edges that carry *larger* information flows about the protected attribute reduces bias at the output to a *greater* extent. This demonstrates that $M$-information flow *can* meaningfully suggest targets for interventions, answering the title's question in the affirmative. We also evaluate bias-accuracy tradeoffs for different intervention strategies, to analyze how one might use estimates of desirable and undesirable information flows (here, accuracy and bias flows) to inform interventions that preserve the former while reducing the latter.

## 1 Introduction

The "reward circuit" of the brain controls much of our behavior [1, 2]. It is believed that addiction is characterized by a strong bias towards immediate rewards in the reward circuit [3], while a different bias, called affective bias, in the same circuit, can lead to depression [4]. Broadly, such biases in the output of the reward circuit control our behavior and responses to stimuli. Recent technological advancements in neural probes and optogenetics [e.g., see 5, 6] are increasingly enabling us to record neural circuits at a high resolution and alter the network by activating or suppressing nodes and links, giving us powerful tools to understand and affect how this circuit processes information. Even a weak understanding of the reward circuit is motivating clinical interventions for individuals suffering from depression, addiction, obsessive compulsive disorder, obesity, etc.; patients are starting to receive surgeries for long-term deep-brain electrode implantation in brain regions that are involved in the reward circuit [7–10]. We identify a new and largely unexplored problem within this context: how do we perform minimal interventions that "correct" undesirable biases without affecting other functions of this important network as much as possible?

In this paper, we propose to use the $M$-information flow framework, a recent advancement of ours, that enables tracking the information flow *about specific messages* within a computational

---

*This work was done while P. Venkatesh and S. Dutta were Ph.D. students at Carnegie Mellon University.
†S. Dutta and N. Mehta contributed equally to this work.

35th Conference on Neural Information Processing Systems (NeurIPS 2021).

circuit [11]. We aim to use this framework to identify interventions that can correct undesirable biases within the circuit, while preserving desirable ones. A difficulty that we face, however, is finding a sufficiently comprehensive dataset with simultaneous recordings of the various brain regions involved in the reward circuit. Although advances in experimental techniques suggest that we are on track to have such datasets in the coming years [12, 13], simultaneous multi-area recordings of specific circuits are still rare. To overcome this issue, we use the context of fairness in artificial intelligence [14, 15] to study the above question, which, as we next discuss, provides an excellent analogy to the neuroscientific context.

The problem of reducing biases to improve fairness in a decision-making system is much like that of reducing undesirable biases in the reward circuit: (i) both systems rely on learned associations from stimuli and responses/labels (i.e., training data); (ii) both systems have an intended objective (to learn a desired association between the features and the true labels, or healthy reward-based learning); and (iii) both systems often learn undesirable associations (e.g., racial or gender bias in artificial neural networks, and biases that cause addiction, depression, or obsessive compulsive disorder in the reward circuit). This suggests that if we want to understand how information flows can inform interventions to correct biases in the reward network, we can simulate such experiments by examining the relationship between information flows and interventions in artificial neural networks (ANNs) trained on biased datasets.

A distinct advantage offered by our $M$-information flow framework [11] is that it can track the flows of *multiple* messages in a neural circuit. In this paper, we adapt our information flow measure to ANNs, and evaluate this measure in an empirical setting involving multiple messages (one each pertaining to bias and accuracy) for the first time. We then perform interventions using different strategies to prune nodes or edges of the ANN, and measure the effect of each intervention by quantifying the change in accuracy or bias at the output of the network.

**Goals and Contributions.** Concretely, the main goals and contributions of this paper are:

1. Our primary goal is to study *whether* measuring information flows about a message can suggest targets for interventions in an ANN, to change how its output is affected by that message. Towards this, our first contribution is in adapting $M$-information flow to the ANN context, developing a meaningful way to quantify it.
2. Secondly, we wish to understand the correlation between the *magnitude* of information flow on an edge and the degree to which an intervention on that edge affects the output. For this, our contribution is an empirical examination of information flows in an ANN, showing that they can predict the effect of interventions in a fairness context.
3. Lastly, we wish to understand how we can use information flows corresponding to two different messages, to remove undesirable behaviors while preserving desirable ones. Towards this, we compare different intervention strategies that are informed by information flows, and compare the bias-accuracy tradeoffs achieved by each of these strategies.

The rest of the paper is organized as follows: we discuss related work below, following which we summarize the fundamentals of the $M$-information flow framework in Section 2. We also show how $M$-information flow can be adapted to ANNs, discuss the setup of the fairness context, and explain the rationale for adopting an empirical approach. In Section 3, we describe how we empirically evaluate our goals: specifically, we discuss the process of estimating $M$-information flow and the design of different intervention strategies. Finally, we present our results on synthetic and real datasets in Section 4, and conclude with a discussion of the implications of our results in Section 5.

**Related work.** The idea of using information flow to understand an ANN (and neural circuits more broadly), could be interpreted as a new type of "explanation" of ANNs. A large number of works have dealt with problems in the fields of explainability, transparency and fairness in machine learning broadly, as well as for ANNs specifically. Molnar [16] provides a good summary of the different approaches taken by many of these methods for explainability. Most of these approaches seek to understand the contribution of individual features [17–20] or individual data points [21, 22] to the output. For ANNs specifically, these can also take the form of visualizations to describe what features or what abstract concepts an ANN has learned [23, 24]. There have also been a number of information-theoretic approaches for measuring bias and promoting fairness in AI systems [25–29].

Our approach in this paper is quite different from these prior works: we want to understand what it is about the *network structure itself* that leads to a certain output. In this sense, our work is similar

in spirit to the idea of "lottery tickets" [30], although we take a very different approach involving information flows. In particular, we want to understand which edges carry information relevant to classification, as well as information resulting in bias, to the output. We also want to know which edges need to be changed in order to produce a desired output, e.g., fairness towards a protected group with minimal loss of accuracy.

Although we examine how information flow can be used to obtain a bias-accuracy tradeoff in a fairness context, our goal is *not* to find *optimal* tradeoffs between fairness and accuracy, which is the focus of many recent works [31–38]. Approaches that directly seek to optimize this tradeoff (e.g., by including bias in the training objective, or by using adversarial methods [39, 40]) will likely perform much better than our method, which uses a much more indirect approach to inducing fairness in an ANN. However, those methods would not tell us how to *edit* the network towards a particular goal, which is what is needed in clinical and neuroscientific application domains. We use the fairness context only as a concrete setting in which to understand whether $M$-information flow can inform interventions[1] in a network with complex flows involving multiple messages.

## 2 Background and Problem Statement

In this section, we provide a brief introduction to our $M$-information flow framework [11], and show how it can be adapted and reinterpreted for ANNs. Then, we set up the fairness problem, and describe how the adapted information flow measure is related to commonly used measures of bias against a protected class. We also discuss why providing theoretical guarantees might be challenging and motivate the need for an empirical approach.

### 2.1 Adapting and Reinterpreting $M$-Information Flow for ANNs

Our $M$-information flow framework provides a concrete way to *define* information flow about a random variable $M$—called the *message*—in a general computational system. By changing what $M$ represents, we can understand the flows of any number of different messages (e.g., if $Y$ and $Z$ are two different variables, we can examine $Y$-information flows and $Z$-information flows in the same system). The computational system, which is modeled after the brain, is a graph consisting of nodes that compute functions, and edges that transmit the results of these computations between nodes. The definition of $M$-information flow satisfies an important property: it guarantees the ability to *track* how information about the message $M$ flows from the input to the output of a feedforward neural circuit.[2] Such a model is completely compatible with a feedforward neural network, where the neurons of the ANN act as the nodes, and the weights connecting neurons from different layers act as the edges of the computational system.

A key feature of our computational system model is how it accounts for the inherent stochasticity of the brain as well as its inputs: nodes can generate noise intrinsically, and the transmissions on the edges are considered to be random variables. But in a trained ANN, the computation at every node is deterministic, specified completely by the weights on the incoming edges and the neuron's activation function.[3] However, we can still continue to think of the edges' transmissions as being random variables because the *input data* for the neural network comes from a distribution (which could also be an *empirical* distribution, i.e., a dataset). We are now in a position to state what our original $M$-information flow definition [11] looks like in the context of ANNs, before proceeding to adapt it for our purposes in this paper.

**Definition 1** (Original $M$-information flow [11]). *Let an arbitrary edge of the neural network at layer $t$ be denoted $E_t$ and let the "transmission" on this edge be denoted $X(E_t)$. Similarly, let a subset of edges at layer $t$ be denoted $\mathscr{E}'_t$ and the set of transmissions on this subset be denoted $X(\mathscr{E}'_t)$. Then we say that information about $M$ flows on the edge $E_t$, i.e., edge $E_t$ has $M$-information flow, if*

$$\exists \, \mathscr{E}'_t \subseteq \mathscr{E}_t \quad s.t. \quad I\big(M; X(E_t) \,\big|\, X(\mathscr{E}'_t)\big) > 0, \tag{1}$$

---

[1]The word "interventions" in this paper refers to *editing* the weights of an ANN, which may correspond to using *targeted* external stimulation in the brain, e.g. through long-term potentiation [41]. We do not mean interventions in the sense of diffuse medications which have a more widespread impact on brain function.

[2]In the original framework, which we designed for the neuroscientific context, we considered the neural circuit as being "feedforward in *time*": i.e., the computational graph is *time-unrolled* in such a way that edges send transmissions from nodes at time $t$ to nodes at time $t + 1$.

[3]We avoid constructions that introduce stochasticity within an ANN for the sake of simplicity.

*where $\mathcal{E}_t$ is the set of all edges in layer $t$, $X(\mathcal{E}_t')$ represents the set of transmissions on $\mathcal{E}_t'$, and $I(M; A \mid B)$ is the conditional Shannon-mutual information between $M$ and $A$, given $B$ [42, Ch. 2].*

The motivation behind this definition is two-fold: (i) One intuitively expects an edge $E_t$ to carry information flow about $M$ if its transmission $X(E_t)$ *depends* on $M$, i.e., if $I(M; X(E_t)) > 0$. Definition 1 satisfies this requirement. (ii) However, it is possible for two edges $E_t^1$ and $E_t^2$ to carry information about $M$ jointly in such a way that $I(M; X(E_t^1)) = I(M; X(E_t^2)) = 0$, but $I(M; \{X(E_t^1), X(E_t^2)\}) > 0$.[4] Therefore, Definition 1 is designed to assign $M$-information flow to both $E_t^1$ and $E_t^2$ in this case: we previously showed [11, 43] that such an assignment is imperative for consistently *tracking* the information flow about $M$ in a computational system.

To adapt this definition of information flow to ANNs, we start by recognizing that our computational system model allowed each outgoing edge of a given node to carry a different transmission. In an ANN, however, the outgoing edges of a given neuron all carry the *same activation*, but with different weights. Consequently, the random variables representing the transmissions are all scaled versions of each other, and have precisely the same information content. Therefore, we can define information flows for the *activations of every node*, rather than for the transmissions of every edge. We then use the properties of edges (specifically, their weights) to construct a definition of information flow for edges, so as to identify the most important outgoing edges of each node and intervene in a more selective manner. Furthermore, Definition 1 only specifies *whether or not* a given edge has $M$-information flow. However, we require a *quantification* of $M$-information flow that will let us compare different nodes or edges, and decide which ones to intervene upon. Keeping these aspects in mind, we propose $M$-information flow for the nodes of an ANN, followed by a quantification of $M$-information flow, and finally a weighted version that assigns different flows to each outgoing edge of a given node:

**Definition 2** ($M$-information flow for ANNs). *Let an arbitrary node of the neural network at layer $t$ be denoted $V_t$, and let the activations of this node by represented by the random variable $X(V_t)$. Similarly, let an arbitrary subset of nodes at layer $t$ be denoted $\mathcal{V}_t'$ and the set of activations of this subset be denoted $X(\mathcal{V}_t')$. Then, we say that the node $V_t$ has M-information flow if*

$$\exists \, \mathcal{V}_t' \quad s.t. \quad I\big(M; X(V_t) \,\big|\, X(\mathcal{V}_t')\big) > 0. \tag{2}$$

*We quantify $M$-information flow by taking a maximum[5] over all subsets of nodes $\mathcal{V}_t'$ in layer $t$:*

$$\mathscr{F}_M(V_t) := \max_{\mathcal{V}_t'} I\big(M; X(V_t) \,\big|\, X(\mathcal{V}_t')\big). \tag{3}$$

*Finally, if $E_t$ is an outgoing edge of the node $V_t$ that has weight $w(E_t)$, then we define the weighted $M$-information flow on that edge as[6]*

$$\mathscr{F}_M(E_t) := |w(E_t)| \, \mathscr{F}_M(V_t). \tag{4}$$

## 2.2   Fairness Problem Setup

The central problem in the field of fair machine learning is to understand how we can train models for classification or regression without learning biases present in training data. Recent examples in the literature have shown why algorithms biased against a protected group can be of great concern [15, 46], with the rise of automated algorithms in application domains such as hiring [15], criminal recidivism prediction [47] and predictive policing [48].

We consider the problem of training artificial neural networks for classification using datasets that have bias in their features and/or labels. The dependencies between the protected attribute (e.g., race, gender, nationality, etc.), the true labels, and the features may be described using a graphical model as shown in Figure 1. We assume that the protected attribute $Z$ influences the features $\{X_i\}$, and possibly the label $Y$, along with some other latent factors encoded in $U$. We then train an ANN using the labels and features to acquire the predicted label $\widehat{Y}$.

---

[4]Simply take $M, X(E_t^2) \sim$ i.i.d. Ber$(1/2)$ and $X(E_t^1) = M \oplus X(E_t^2)$, where $\oplus$ represents the exclusive-OR operation between two binary variables.

[5]The intuition behind using a maximum comes from a notion called "synergy" [44, 45]. Essentially, if $I(M; A \mid B) > I(M; A)$, then $B$ is said to contribute *synergistic information* about $M$ to $A$. By taking a maximum, we attempt to include synergistic contributions from all other activations in the same layer.

[6]Our definition of weighted $M$-information flow for the edges of the ANN is admittedly heuristic, however it has a simple rationale: an edge with a larger weight has a greater influence on the activation of a target node as compared to an edge with a smaller weight, given the same input to both.

Recall that our goal is to measure the information flows of two different messages: (i) information flows about the protected attribute, i.e., $Z$-information flows, which we also refer to as *bias flows*; and (ii) information flows about the true label, i.e., $Y$-information flows, which we also refer to as *accuracy flows*, as these are responsible for accuracy at the output. The measure of bias we consider at the output is an information theoretic version of statistical (or demographic) parity [49], which has also been used in many previous works [36]. This is because $Z$-information flow at the output is simply $I(Z; \widehat{Y})$, since there are no other edges to condition upon.[7]

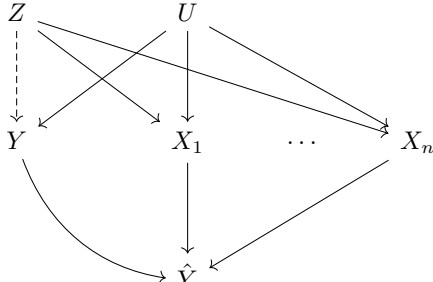

Figure 1: A graphical model representing the causal relationships assumed between the variables used in the fairness problem. The ANN's output, $\widehat{Y}$, depends on the features $\{X_i\}$ and the true labels $Y$, which in turn may be influenced by the protected attribute $Z$ and some latent variables $U$ (where $Z \perp\!\!\!\perp U$). Here, $U$ encodes latent factors responsible for "accuracy", i.e., the relationships between $\{X_i\}$ and $Y$ that we *want* to capture. The dashed line from $Z$ to $Y$ indicates that the true labels may or may not be biased (e.g., we assume $Y$ is unbiased in our synthetic dataset, but real datasets will likely have bias).

## 2.3 Theoretical Counterexamples on $M$-Information Flow and Interventions

This section explains why we adopt an empirical approach in this work, and why our results might be considered non-obvious. Intuitively, we might expect that intervening on an edge with a large $M$-information flow (e.g., by deleting it) will have a large effect on the dependence between $\widehat{Y}$ and $M$. However, there exist counterexamples showing that both the above statement and its converse do not always hold true.

**Counterexample 1** (*$M$-information flow does not imply interventional effect*)**.** Consider an ANN that generates an intermediate feature at some node $V_t$ which is strongly correlated with $Z$, and thus has a large $Z$-information flow. However, this feature need not contribute to $\widehat{Y}$: for example, all weights leading out of $V_t$ could be zero. In such a case, pruning $V_t$ will have no effect on the bias at the output, even though $\mathcal{F}_Z(V_t)$ can be arbitrarily large. □

**Counterexample 2** (*Absence of $M$-information flow does not imply absence of interventional effect*)**.** Consider an ANN (and a suitable data distribution) in which the flow of a binary variable $M \in \{0, 1\}$ to $\widehat{Y}$ is "gated" by an independent random variable $W$. More precisely, suppose the activation of a node $V_t$ is a variable $W$ that satisfies $\mathcal{F}_M(V_t) = 0$. Further, let a different node $V_{t+1}$ (say with ReLU activation) receive both $M$ and $W$ as inputs with equal weights. If $W = 0$, then $X(V_{t+1})$ will depend on the value of $M$; on the other hand if $W < -1$, then $X(V_{t+1})$ will always be zero, and hence independent of $M$. If $\widehat{Y} = X(V_{t+1})$, then pruning the node $V_t$ (i.e., intervening on $W$) can change how $\widehat{Y}$ depends on $M$, even though $V_t$ itself has no $M$-information flow. □

The title of the paper asks a non-obvious question for yet another reason: it asks whether an *observational* measure of information flow can predict the effect of *interventions*. The field of Causal Inference [51] shows that it is impossible, *in general*, to predict causal effects using purely observational methods (i.e., by observing empirical data from a joint distribution of random variables, without manually perturbing them in some way). This fact, along with the aforementioned counterexamples, illustrates why it is difficult to provide theoretical guarantees about whether information flows can predict interventions (we also explored this issue in [43]). This is why we adopt an empirical approach in this paper, to examine whether such predictions can be made in *practice*.

## 3 Methods: Estimating Information Flow and Implementing Interventions

In this section, we discuss how we estimate the information flow measure described earlier, and propose a few different intervention strategies for "editing" a trained neural network.

---

[7]It is also straightforward to extend our work to alternative definitions of bias such as Equalized Odds [28, 50] by suitably modifying Definition 2. However, it is not clear if such a measure can still be meaningfully interpreted as an "information flow".

### 3.1 Estimating Information Flow

Conditional mutual information is a notoriously difficult quantity to estimate [52]. However, in our empirical study, we only consider datasets where the true label $Y$ and the protected attribute $Z$ are binary, which makes estimation considerably easier.[8] We use a classification-based method to estimate the conditional mutual information that appears in Definition 2.

First, we construct separate classifiers for predicting $Y$ and $Z$ from the intermediate activations of a node (or a subset of nodes) in a chosen layer, which we call $X^{\text{int}}$. The generalization accuracy of this classifier indirectly tells us the extent to which information about $Z$ (say) is present in $X^{\text{int}}$. More precisely, if the generalization accuracy of classifying $Z$ from $X^{\text{int}}$ is $a$, that means the expected probability of error in correctly guessing $Z$ from $X^{\text{int}}$ is $P_e := 1 - a$. Then, from Fano's inequality [42, Ch. 2], we have:

$$H(Z \mid X) \leq H_b(P_e) + P_e \log(|\mathcal{Z}| - 1), \qquad (5)$$

where $H_b$ is the binary entropy function, and $\mathcal{Z}$ is the sample space of $Z$. Since $Z$ is binary, $\mathcal{Z} \in \{0, 1\}$, hence $|\mathcal{Z}| = 2$. If we further assume $Z \sim \text{Ber}(1/2)$, then $H(Z) = 1$ bit. This simplifies the above equation to:

$$H(Z \mid X^{\text{int}}) \leq H_b(P_e) = H_b(1 - a) \qquad (6)$$

$$\Rightarrow \; I(Z; X^{\text{int}}) = H(Z) - H(Z \mid X^{\text{int}}) = 1 - H(Z \mid X^{\text{int}}) \geq 1 - H_b(1 - a). \qquad (7)$$

Therefore, given any classifier that can predict $Z$ from $X^{\text{int}}$ with generalization accuracy $a$, we can compute a lower bound on the mutual information between $Z$ and $X^{\text{int}}$, with a better classifier providing a tighter lower bound. This allows us to compute all conditional mutual information quantities required by Definition 2 using the chain rule [42, Ch. 2]:

$$I(Z; X_1^{\text{int}} \mid X_2^{\text{int}}) = I(Z; X_1^{\text{int}}, X_2^{\text{int}}) - I(Z; X_2^{\text{int}}), \qquad (8)$$

where $X_1^{\text{int}}$ and $X_2^{\text{int}}$ are the activations of two intermediate nodes (or subsets of intermediate nodes) in the ANN. We use both linear and kernel Support Vector Machines for our classifiers. Further details on estimating the mutual information can be found in Appendix A in the supplementary material.

### 3.2 Intervention strategies

We perform interventions on an ANN by pruning its edges. To evaluate bias-accuracy tradeoffs, we use "soft-pruning" of edges, i.e, gradually reducing their weights rather than removing them outright. We compare a number of pruning strategies, which differ based on the following factors:

1. **Pruning method.** We consider three methods: (i) pruning entire *nodes*; (ii) pruning individual *edges*; and (iii) pruning *paths* from the input layer to the output layer. Pruning nodes, i.e., simultaneously decreasing the weights of all outputs of a particular node, can be interpreted as decreasing the dependence of future computations on the derived feature that this node represents. Pruning individual edges has the advantage of being extremely specific, potentially allowing for similar results with minimal changes to the network. Pruning paths can be interpreted as stopping the largest net flow of information from the input features all the way to the output.

2. **Pruning metric.** For each of the above methods, we use two different metrics: the primary metric is based on the ratio of bias flow to accuracy flow, and a secondary metric is the ratio of accuracy flow to bias flow, which we use as a "control" to show that our tradeoff results are not merely a product of chance. When soft-pruning nodes, the primary metric we use is the ratio of flows at the node, i.e. $\mathscr{F}_Z(V_t)/\mathscr{F}_Y(V_t)$. When soft-pruning edges, our primary metric is the *weighted* ratio of flows on the edge, i.e., $|w(E_t)| \, \mathscr{F}_Z(E_t)/\mathscr{F}_Y(E_t)$ (note that we need to re-weight the ratio since weights in $\mathscr{F}_Z(E_t)$ and $\mathscr{F}_Y(E_t)$ would otherwise cancel). Finally, when soft-pruning paths, we score each path using the smallest edge-metric, and choose the path $P$ having the largest score:

$$P^* = \underset{P}{\arg\max} \; \min_{E_t \in P} \; |w(E_t)| \cdot \mathscr{F}_Z(E_t) \, / \, \mathscr{F}_Y(E_t). \qquad (9)$$

3. **Pruning level.** In each of the above cases, we soft-prune the top "$k$" nodes, edges or paths, which exhibit the largest value of the respective pruning metric. Pruning too much will likely hinder all flows, so we test a few different values of $k$ to find optimal "level" of pruning.

---

[8]Extensions to non-binary $M$ (e.g., intersectional attributes) are certainly possible, and would require the use of different mutual information estimators, suitably tailored to the dataset [53–57].

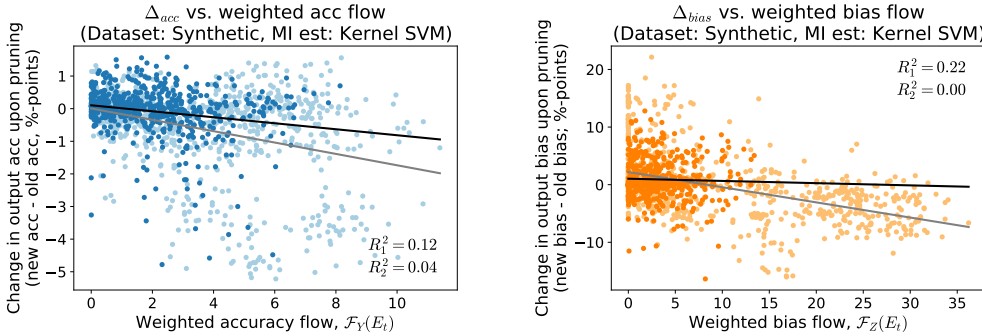

Figure 2: (Left) A plot showing the dependence between the weighted $Y$-information flow of every edge and the change in accuracy ($\Delta_{acc}$) at the output of the ANN upon pruning that edge, for the synthetic dataset. (Right) The same for weighted $Z$-information flow and change in bias ($\Delta_{bias}$). In both figures, as the information flow of an edge increases, there is a greater decrease in the accuracy or bias at the output upon pruning that edge. The lighter data points indicate edges in the first layer, while the darker points represent edges in the second (final) layer, with grey and black lines representing linear fits. $R_i^2$ is the fraction of variance explained by the linear fit, for points in layer $i$.

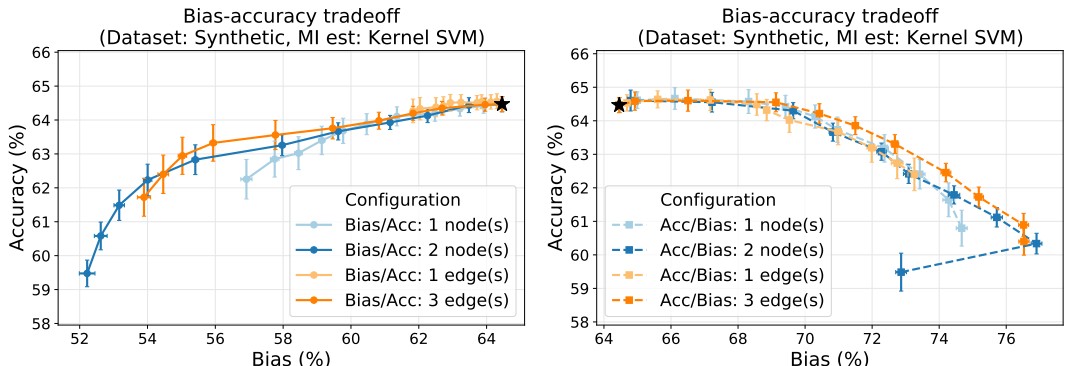

Figure 3: Figures showing the tradeoff between fairness and accuracy when gradually pruning nodes or edges of an ANN trained on the synthetic dataset. ★ denotes the accuracy and bias after training but before pruning. The legend indicates the pruning strategy used: pruning based on bias-to-accuracy flow ratio (left) or accuracy-to-bias flow ratio (right); pruning nodes or edges; and the number of nodes or edges pruned. Error bars represent one standard error of the mean. Pruning based on the bias-to-accuracy ratios causes bias to fall faster than accuracy, while pruning based on accuracy-to-bias ratios causes bias to *increase* while accuracy falls.

## 4 Results

### 4.1 Synthetic Dataset

First, we examine information flows for a small neural network trained on a synthetic dataset. The synthetic dataset is generated in a manner similar to Figure 1 (see Appendix B.1 in the supplementary material for details). The dataset has three continuous-valued features $X_1$, $X_2$ and $X_3$, a binary label $Y$, and a binary protected attribute $Z$. $X_1$ and $X_2$ are designed to receive a large causal influence from $Z$, while $X_3$ and the labels $Y$ have no influence from $Z$. Lastly, all three features independently provide information about $Y$, i.e., they are noisily correlated with $Y$, with independent noise terms.

For simplicity, the neural network was taken to have just one hidden layer with three neurons, with leaky ReLU activations. The output layer is a one-hot encoding of the binary $\widehat{Y}$ and cross-entropy loss was used for training. The data was split, with 50% used for training the neural network, and 50% for estimating information flows on the trained network. We used a kernel SVM classifier to estimate information flow (as described in Section 3.1) and nested cross validation to fit SVM hyperparameters and estimate generalization accuracy. Finally, all analyses are repeated across 100

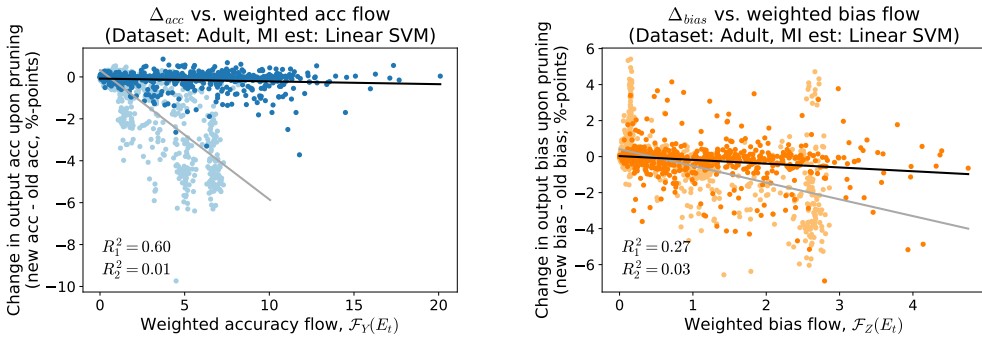

Figure 4: Plots analogous to those in Figure 2, for the Adult dataset trained on the smaller ANN.

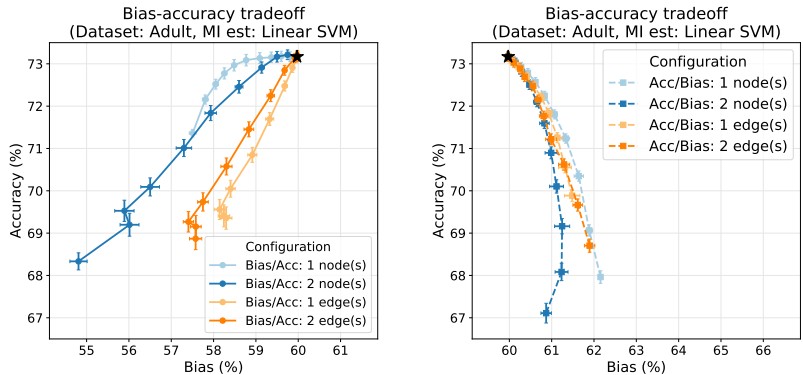

Figure 5: Tradeoff plots analogous to Figure 3, for the Adult dataset trained on the smaller ANN.

neural networks trained on the same data but with different random weight initializations. Further details are in Appendix C.1 in the supplementary material.

We first analyzed whether the extent of change in output bias ($\Delta_{bias}$) or accuracy ($\Delta_{acc}$) upon pruning an edge is related to the magnitude of $Z$- or $Y$-information flow on that edge. We completely pruned each edge of the trained network, keeping all other edges intact, and examined the change in accuracy and bias at the output (see Figure 2). The results show that pruning edges with larger magnitudes of weighted accuracy or bias flow tends to produce larger reductions in accuracy or bias at the output, respectively. We also see a marked difference in the slopes of the black and grey lines, indicating that edges from the second layer are less likely to change accuracy or bias upon being pruned: this is discussed in the following subsection. While the linear fits do not capture a large fraction of the overall variance, the *dependence* between information flow magnitude and $\Delta_{bias}$ or $\Delta_{acc}$ is evident: for completeness, correlation values and statistical significance tests are provided in Appendix D in the supplementary material.

Next, we analyzed how bias and accuracy evolve upon "soft-pruning" the edges gradually, for different pruning strategies outlined in Section 3.2. Figure 3 shows that, when pruning on the basis of bias-to-accuracy flow ratio, bias initially falls much faster than accuracy: on average, bias can be reduced almost 10 percentage points, at a cost of just 2 percentage points in accuracy. The tradeoff curves produced by different pruning methods and levels were similar for this dataset, but interestingly, pruning on the basis of the accuracy-to-bias flow ratio causes accuracy to remain steady before falling, while bias *increases*. This may suggest the presence of some redundancy: when edges most responsible for accuracy are pruned, other edges (presumably from nodes that have larger bias) are able to compensate, but at the cost of increasing bias.

## 4.2 Real Datasets: Modified Adult and MNIST

We also performed the same analyses on the Adult dataset from the UCI machine learning repository [58] and the MNIST dataset [59]. The Adult dataset, which comes from the 1994 US census, consists of a mix of numerical and categorical features for classifying people with annual incomes

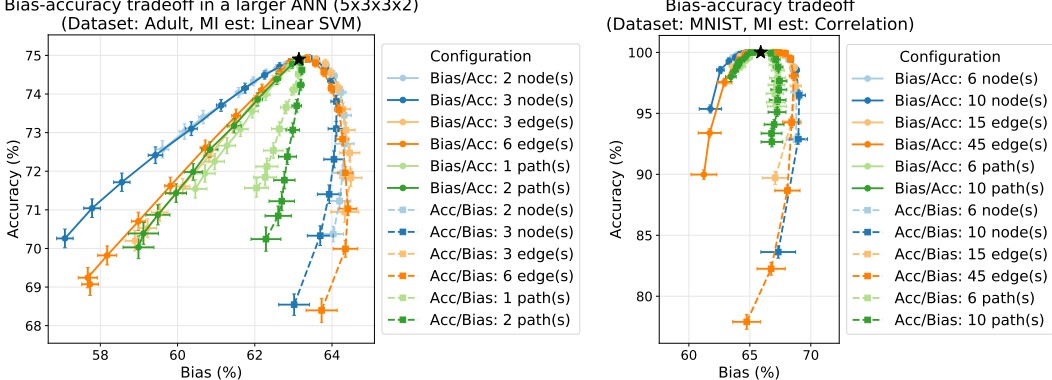

(a) Tradeoff plot for the modified Adult dataset, trained on a slightly larger ANN, with five input features and two hidden layers with three neurons each.

(b) Tradeoff plot for the MNIST dataset, as described in Appendix B.3 and C.1.

Figure 6: Tradeoff plots for the Adult and MNIST datasets. In both plots, the green lines are for the path-based pruning method (described in Section 3.2). Note that both the bias-to-accuracy and accuracy-to-bias flows have been shown in the same plot.

less than and greater than $50k. We perform our experiments on two ANNs: a small neural network, identical to the one used for the synthetic dataset, and a slightly larger network having five input features, two hidden layers with three neurons each, and a one-hot-encoded output layer. For the smaller network, we use three numerical features—`education-num`, `hours-per-week` and `age`—and take `sex` to be the protected attribute. For the larger network, we additionally use continuous-valued embeddings derived from two categorical features, `occupation` and `workclass`, to test that our observations are not limited to a specific topology. We also used only a subset of the records in order to equalize the number of individuals with high and low incomes and the number of male and female individuals. However, we introduced a bias in the true labels by skewing the dataset towards higher incomes among males and lower incomes among females (at a ratio of 2:1). In the figures, we simply use "Adult dataset" to refer to this modified version of the Adult dataset. Information estimates were performed using a linear SVM (refer to Section 3.1).

Results analogous to the synthetic dataset are shown in Figures 4 and 5 for the smaller ANN, and in Figure 6a for the larger ANN trained on the modified Adult dataset. These results reflect the trends seen in the synthetic dataset. The smaller ANN shows a stronger dependence between information flow and interventional effect for the Adult dataset than when trained on the synthetic dataset (compare Figures 2 and 4). This might be because the synthetic data had two *highly* biased features, whereas the modified Adult dataset likely has much less bias in its features. Figures 2 and 4 also show that interventions in the first layer (lighter points) tend to have a greater effect on output accuracy/bias than those in the second layer (darker points)—this is particularly pronounced in Figure 4 (left). We believe this is because of the intrinsic redundancy in one-hot encoding: pruning a single edge in the second layer will only affect *one* of the two output neurons, which encode the binary random variable $\widehat{Y}$, and therefore provide the same information.

In Figures 5 and 6a, pruning edges does worse than pruning nodes (e.g., compare 2-nodes with 6-edges in Figure 6a, both of which prune 6 edges). This suggests that reducing the impact of an entire (derived) feature at a node is a more robust intervention than attempting to edit individual edges. We also see that when pruning using accuracy-to-bias flow ratios (dashed lines), accuracy falls quickly and bias increases only modestly (in contrast to Figure 3 for the synthetic dataset). We believe this indicates that the modified Adult dataset is dominated more by accuracy flows while the synthetic dataset is dominated by bias flows (this is also substantiated by the magnitudes of information flow seen in Figures 2 and 4). Finally, the path-based pruning strategy (which makes more sense for deeper networks) is shown in Figure 6a, and is similar to the edge-based methods in its tradeoff performance when using weighted bias-to-accuracy flow ratios. Interestingly however, when pruning based on weighted accuracy-to-bias flow ratios, there is no increase in bias, suggesting that path-based methods might have greater specificity to a particular message.

To test the scalability of our methods to wider and deeper ANNs, we also repeated our analyses on an ANN with 5 hidden layers consisting of 6 units each, trained on the MNIST dataset (details are provided in Appendix B.3 and C.1 in the supplementary material). Information flow was estimated using a fast correlation-based estimator (see Appendix A.3), and the results mirror the trends from the other datasets (see Figure 6b). An interesting observation is that, as the size of the network increases, one has to prune a larger number of nodes, edges or paths to achieve similar trade-offs. We also performed a dependency analysis (not depicted), similar to that of Figures 2 and 4. However, these showed a very poor linear fit; correlation values were also low with inconsistent statistical significance (see Appendix D for details). These observations might be attributed to the fact that pruning individual edges is unlikely to have a significant impact on the output in such a large network.

Further results, including a more extensive sampling of pruning strategies and fast correlation-based information flow estimates, are provided in the supplementary material. Code to generate these results is available online at `https://github.com/praveenv253/ann-info-flow`.

## 5  Discussion

Our results show that $M$-information flow can indeed suggest targets for intervention in a trained artificial neural network, to change its behavior at the output. We can simultaneously measure the information flows of multiple variables within the system, and make edits to prevent undesirable behaviors while preserving desirable ones.

This has important implications for neuroscience and clinical practice, where an observational understanding appears to be necessary prior to deciding the form of targeted intervention. Today, studies typically report results from interventions at a small number of sites (usually just one), while the number of sites for *potential* interventions is large. The number of possible interventions grows exponentially with the number of sites, which, along with the amount of data required for each, and the irreversibility of interventions, necessitates the use of new approaches that do not iterate through candidate interventions. On the other hand, there are several well-established neural recording modalities for making *observations*, and there have been rapid advances in non-invasive recording and stimulation techniques [60–65]. Thus, neuroscientists who wish to explore causal mechanisms in neural circuits can measure information flows to narrow down target brain areas for confirmatory experiments that use optogenetic stimulation—in this way, information flows can inform experiment design. Developing such an understanding is crucial for the eventual goal of treating brain disorders (e.g., in the reward circuit); furthermore, mapping information flows might provide a roadmap to personalized deep-brain stimulation strategies [66].

Our work is intended to be an initial exploration of how we might make interventions by studying information flows. Our results show that there is substantial room for improvement: there is a large variance in how well the interventions suggested by our approach perform (note that this is not entirely unexpected, as discussed in Section 2.3). Future studies may examine other intervention strategies, datasets, and more sophisticated networks. Some challenges remain in extending these results to a neuroscientific setting. Firstly, the computational system model (nodes and edges) [11] may be too simplistic for biological neurons where even single cell has sophisticated dynamics. Secondly, measurements in real neural systems are frequently noisy, corrupted by background brain activity as well as sensor noise, while we assume here that noiseless measurements can be made. Similarly, although technologies for making interventions on specific cells are rapidly advancing [6], it is still difficult to perform precise neuromodulation, especially without genetic manipulation of cells.

**Potential negative societal impacts.** This study discusses approaches for making informed interventions in neural circuits, with potential applications to the brain. As with any new technology, this carries potential risks of misuse or abuse, e.g., imprudent or overzealous interventions in clinical or non-clinical settings, with or without the subject's consent. Proper regulation and oversight will be key in preventing such misuse. On a more imminent time-scale, the techniques developed here might be misused in fairness settings, despite having no formal guarantees. Although our paper uses the context of fairness to understand whether information flows can suggest interventions, we have not tested this approach rigorously enough for it to be employed to induce fairness in real systems—indeed, this was not our goal. While we believe that information flow analyses may, in future, be used to guide the systematic editing of neural networks for purposes such as bias reduction, that would require a more thorough analysis of optimal information measures and pruning strategies.

## Acknowledgments

We thank Cosma R. Shalizi for suggesting a comparison between the magnitudes of information flow and interventional effect. We also thank Robert E. Kass and José M. F. Moura for useful discussions.

**Funding disclosures.** This material is based upon work supported by the National Science Foundation under Grant No. CCF-1763561. P. Venkatesh was supported in part by a Fellowship in Digital Health from the Center for Machine Learning and Health at Carnegie Mellon University. S. Dutta was supported in part by a Cylab Presidential Fellowship and a K&L Gates Presidential Fellowship. N. Mehta was funded in part by a fellowship supported by the CMU-Portugal program.

**Disclaimer.** This paper was prepared for informational purposes and is not a product of the Research Department of J.P. Morgan. J.P. Morgan makes no representation and warranty whatsoever and disclaims all liability, for the completeness, accuracy or reliability of the information contained herein. This document is not intended as investment research or investment advice, or a recommendation, offer or solicitation for the purchase or sale of any security, financial instrument, financial product or service, or to be used in any way for evaluating the merits of participating in any transaction, and shall not constitute a solicitation under any jurisdiction or to any person, if such solicitation under such jurisdiction or to such person would be unlawful.

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
