# Can Information Flows Suggest Targets for Interventions in Neural Circuits?
## Appendices

Praveen Venkatesh, Sanghamitra Dutta, Neil Mehta and Pulkit Grover

**A Note on Error Bars**

Error bars in all tradeoff plots in the paper represent one standard error of the mean (i.e., roughly corresponding to a 68%-confidence interval on the mean), both for accuracy and bias. The mean accuracy and bias at different pruning levels was computed on 100 runs of the whole analysis, each starting with a different weight initialization for the ANN.

## A  Additional Details on Estimating Information Flow

### A.1  Details on Classifiers for Estimating Mutual Information

In this section, we describe what classifiers were used when $M$-information flow was estimated as described in Section 3.1. We tried two different classifiers: a linear support vector machine (SVM) and a kernel SVM (using a radial basis function kernel, which was approximated using a Nyström kernel approximation and optimized using stochastic gradient descent).

We examined the performance of both these classifiers on the synthetic as well as the modified Adult datasets. However, we found that linear SVM did very poorly on the synthetic dataset as a result of the intrinsic non-linearity present in its design (as will be described in Appendix B.1, which appears shortly). On the other hand, we found that both classifiers performed well on the modified Adult dataset: in particular, linear SVM—being a simpler classifier with fewer hyperparameters—had much lower *variance* than kernel SVM, and was much faster to fit, prompting us to utilize only linear SVM for the modified Adult dataset.

For the kernel SVM, the Nyström approximation was taken to have 100 components. At any given stage, the inputs to the classifiers were standardized as part of a pipeline before being running SVM. All hyperparameters—$C$ for linear SVM, and $(C, \gamma)$ for kernel SVM—were drawn from log-uniform distributions with a range of $[10^{-2}, 10^2]$. We used nested cross-validation, optimizing hyperparameters in the inner loop (4 folds), and producing our final information estimates in the outer loop (5 folds). We optimized hyperparameters using a randomized search with 25 parameter draws, both for linear and kernel SVM. The mutual information estimate used the average generalization accuracy estimated on the test data of each of the 5 folds in the outer loop.

All of the aforementioned classifiers were implemented using Scikit-Learn [67] in Python.

### A.2  Keeping Information Estimates Positive

In some cases, we may find that the estimate for $I(Z; (X_1^{\text{int}}, X_2^{\text{int}}))$ is smaller than that for $I(Z; X_2^{\text{int}})$; this is because our estimates are lower bounds. Although adding variables can never decrease mutual information, in practice, adding features may reduce the accuracy of a classifier, especially if the extra features only contribute noise and do not help with discriminability.[9] In such cases, we simply truncate the conditional mutual information to zero, to prevent it from becoming negative.

### A.3  A Mutual Information Estimate based on Correlation

In addition to the classifier-based estimate for mutual information presented in Section 3.1, we also consider a simpler measure of mutual and conditional mutual information based on a Gaussian approximation [42, Sec. 8.5]. In this case, the mutual information can be expressed in closed form, in

---

[9]Ideally, adding more features to a classifier should not reduce classification accuracy. However, this is incumbent on having good feature selection mechanisms and sufficient computational resources to estimate optimal hyperparameters in the higher-dimensional feature space. In practice, these constraints imply that features that do not provide useful information decrease our ability to classify based on the other useful features.

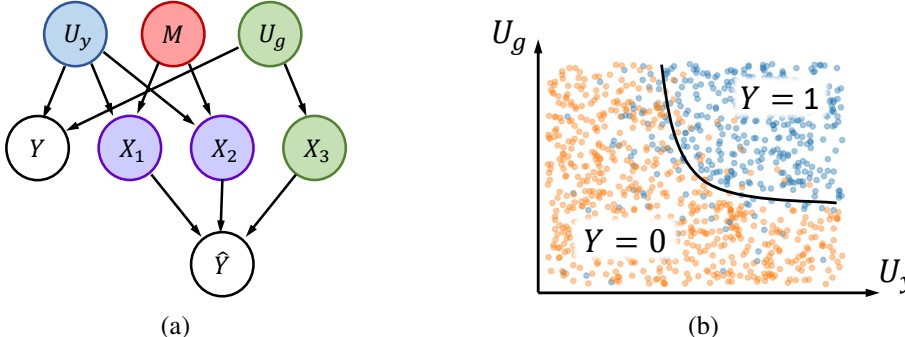

(a)        (b)

Figure 7: A depiction of how the synthetic data was generated. (a) The graph corresponding to the structural equation model used to generate the synthetic dataset. (b) A cartoon representation of the relationship between the latent variables $U_y$ and $U_g$, and the true labels $Y$, shown here for a uniform distribution over $U_y$ and $U_g$ for clarity. In the actual dataset, $U_y$ and $U_g$ were drawn from Gaussian distributions.

terms of the joint covariance matrix of $Z$ and $X^{\text{int}}$. For example, if $Z$ and $X^{\text{int}}$ are scalar Gaussian random variables, then

$$I(Z; X^{\text{int}}) = \frac{1}{2} \log_2 \big( 1 - \rho^2(Z, X^{\text{int}}) \big) \text{ bits,} \tag{10}$$

where $\rho(Z, X^{\text{int}})$ is the *correlation* between $Z$ and $X^{\text{int}}$. In general, for computing information flow, we need to compute the conditional mutual information between $Z$ and $X_1^{\text{int}}$ given $X_2^{\text{int}}$. By virtue of the chain rule shown in equation (8), it suffices to find an estimator for the *mutual information*.

Suppose $[Z, X^{\text{int}}]$ comprise a jointly Gaussian random vector with covariance matrix $\Sigma$:

$$\Sigma = \left[ \begin{array}{cc} \Sigma_{ZZ} & \Sigma_{ZX} \\ \Sigma_{XZ} & \Sigma_{XX} \end{array} \right]. \tag{11}$$

We can construct the covariance matrix that would be produced if $Z$ and $X^{\text{int}}$ had the same marginals, but were independent of one another:

$$\widetilde{\Sigma} = \left[ \begin{array}{cc} \Sigma_{ZZ} & 0 \\ 0 & \Sigma_{XX} \end{array} \right]. \tag{12}$$

Then, the mutual information between $Z$ and $X^{\text{int}}$ is given by the KL-divergence between their joint distribution and the product of their marginals, which can be written in closed form for Gaussians:

$$I(Z; X^{\text{int}}) = \frac{1}{2 \log_e(2)} \left( \text{Tr}\{\widetilde{\Sigma}^{-1}\Sigma\} - d + \log_e \frac{\det \widetilde{\Sigma}}{\det \Sigma} \right) \text{ bits,} \tag{13}$$

where Tr and det refer to the trace and determinant of a matrix respectively, while $d$ is the number of elements in the vector $[Z, X^{\text{int}}]$.

This approximation no longer provides a bound on mutual information, but is extremely fast to compute. In Appendix C and D, we compare it with other measures to evaluate its performance.

## B   Details on All Datasets

### B.1   Details on the Synthetic Data Model

The synthetic dataset consisted of 10,000 data points, with 5000 used to train the neural network and 5000 used to estimate information flows. The data was generated according to a structural equation model, whose graph is shown in Figure 7a. We start with two latent variables, $U_y$ and $U_g \sim$ i.i.d. $\mathcal{N}(0, 1)$, and the protected attribute $M \sim \text{Ber}(0.5)$, with $M \perp\!\!\!\perp \{U_y, U_g\}$. We then set $Y$ based on a nonlinearity as shown in Figure 7b. The boundary of the nonlinearity is set to be

$$U_y = \frac{1}{1 + U_g} - 1. \tag{14}$$

If $d(U_y, U_g)$ is the signed distance from any point $(U_y, U_g)$ to this boundary, then

$$Y \mid U_y, U_g \sim \text{Ber}\left(\frac{1}{1 + e^{-3d(U_x, U_y)}}\right). \tag{15}$$

$X_1$, $X_2$ and $X_3$ are designed to indirectly convey information about $Y$, by encoding $U_y$ and $U_g$ in a manner biased by $M$. $X_1$ and $X_2$ are chosen to be biased, with

$$X_1 \mid U_y, M \sim \mathcal{N}(0.7MU_y, 0.04), \tag{16}$$
$$X_2 \mid U_y, M \sim \mathcal{N}(0.5MU_y, 0.04), \tag{17}$$
$$X_3 \mid U_g, M \sim \mathcal{N}(0.1U_g, 0.04). \tag{18}$$

Note that $X_1$ and $X_2$ communicate information about $U_y$ only when $M = 1$. Therefore, these two features are informative for classifying $Y$ only when $M = 1$, and are completely non-informative when $M = 0$, inducing bias at the output. $X_3$, on the other hand, is an unbiased feature, which is equally informative for both $M = 0$ and $M = 1$.

Using the data generation model described above, we first generated 15,000 data points. We then randomly subsampled data points to balance out classes: we picked 2500 data points for each of the true label and protected attribute values $(Y, Z)$ of $(0,0)$, $(0,1)$, $(1,0)$ and $(1,1)$. As a result, there were an equal number of data points in the dataset, for each possible combination of $(Y, Z)$. This had the benefit of imposing independence between the true label $Y$ and the protected attribute $Z$. Further, it ensured that there were an equal number of data points with true label 0 and 1, and an equal number of data points with protected attribute 0 and 1.

## B.2 Details on the Modified Adult Dataset

The modified Adult dataset was generated from the Adult dataset by balancing out the number of male and female individuals, as well as balancing out the number of individuals with high ($\geq$ \$50,000) and low ($<$ \$50,000) incomes. However, in contrast to the balancing process described above for the synthetic dataset, the true label and protected attribute were not made to be independent of each other. This was because balancing all four combinations of $(Y, Z)$ led to a very low bias at the output $\hat{Y}$ of an ANN trained on such a dataset, even in the absence of any fairness considerations.

Instead, we skewed the joint distribution of incomes and genders, so that two-thirds of males had high incomes, while two-thirds of females had low incomes. More precisely, we randomly drew as many data points as possible from the dataset, while ensuring that our sample satisfied:

$$(M, <) : (M, \geq) : (F, <) : (F, \geq) = 1 : 2 : 2 : 1,$$

where $M$, $F$, $\geq$ and $<$ represent the number of males, females, individuals with high incomes and individuals with low incomes in our sample, respectively. We chose this particular skew (as opposed to having two-thirds of females with high incomes, for instance) simply to maximize the total number of data points in our dataset.

After subsampling, we further discarded a few arbitrarily chosen data points in our reduced dataset so that the total number of data points was an integer multiple of 10, which was the minibatch size used in ANN training. This skewed subsampling process (followed by discarding) left us with a total of 10,600 data points in our sample, down from 48,842 data points in the original Adult dataset (when considering both train and test sets). We had 5293 individuals with high incomes and 5307 individuals with low incomes. Incidentally, we also had 5293 female individuals and 5307 male individuals. The exact number of $(M, <)$, $(M, \geq)$, $(F, <)$ and $(F, \geq)$ individuals was 1769, 3538, 3538 and 1755 respectively.

## B.3 Details on the MNIST Dataset

We used a part of the MNIST dataset to test whether our results could be scaled to deeper and wider ANNs. For the dataset, we used four digits in two pairs: (4, 9) and (5, 8). These pairs were chosen based on a review of t-SNE plots of the MNIST dataset found in the literature [e.g., see 68]: clusters corresponding to 4 and 9 were found to be very close to each other, as well as those corresponding to 5 and 8, suggesting the similarity between the two numbers in each pair. Since each pair also consists of one odd and one even number, we took the protected attribute $Z$ to represent even vs. odd digits.

We skewed the training dataset to induce bias in the output, by having more training examples for 9 in the (4, 9) class, and more examples for 8 in the (5, 8) class at a 2:1 ratio. This maintained an equal number of examples of both classes while also having an equal number of odd and even examples overall. However, this skew induced the network to prefer 9 over 4 at the output neuron for $\widehat{Y} = 0$ and to prefer 8 over 5 for the output neuron for $\widehat{Y} = 1$, allowing us to also guess whether the number was odd or even, based on $\widehat{Y}$.

## C   Details on Data Analysis

For all our analyses and datasets, 50% of the entire dataset was used for training the ANN, and the other 50% for analyzing information flow, including measuring the tradeoff performance (e.g., Figure 5) and dependence plots (e.g., Figure 4).

### C.1   ANN training

For the synthetic dataset, $\widehat{Y}$ was estimated using an ANN with one hidden layer consisting of three neurons. The input layer also had three neurons, for each of the three features, and the output layer was a one-hot encoding of the binary output $\widehat{Y}$. The activation functions of all neurons were Leaky ReLU. The input features were not standardized before training the ANN since they were already designed to be of comparable dynamic ranges. Training was performed for 50 epochs with a minibatch size of 10, a momentum of 0.9, and a learning rate of $3 \times 10^{-2}$.

In the case of the modified Adult dataset, we used two different ANN configurations: the first was a smaller ANN identical to the one used for the synthetic dataset, while the second was a slightly larger ANN with two hidden layers, three neurons in each hidden layer and five inputs. The activation functions of all neurons were Leaky ReLU again. For the smaller ANN, we used three numerical features—`education-num`, `hours-per-week` and `age`—and took `sex` to be the protected attribute. Since these features had large disparities in their dynamic ranges, we individually standardized these features before providing them as inputs to the ANN for training. For the larger ANN, we additionally used continuous-valued embeddings derived from two categorical features, `occupation` and `workclass`. For both the smaller and larger ANNs, training was performed for 50 epochs with a minibatch size of 10, a momentum of 0.9, and a learning rate of $3 \times 10^{-3}$.

Finally, for the MNIST dataset, the network configuration was fully connected with Leaky ReLU activation and had layer sizes $[784, 6, 6, 6, 6, 6, 2]$: 784 input features (flattened $28 \times 28$ MNIST images), 2 output neurons, and five hidden layers with 6 neurons each. Training was performed for 50 epochs with a minibatch size of 10, a momentum of 0.9 and a learning rate of $3 \times 10^{-3}$. The trained network was able to achieve approximately 98% accuracy and had a bias of around 65% (accuracy of classifying $Z$ from $\widehat{Y}$). We performed the $M$-information flow analysis for all hidden layers and the output layer; we excluded the input layer for computational tractability. We also used correlation-based estimates for mutual information, as described in Appendix A.3, for faster computation. Using our $M$-information flow measure to estimate the $Y$- and $Z$-information flows, we could once again produce tradeoff plots similar to those in Figures 3, 5 and 6.

We trained 100 different ANNs on the same 50% segment of training data, each with different random weight initializations. The error bars in the tradeoff plots (Figures 3, 5 and 6a) and the points in the dependency plots (Figures 2 and 4) are over these 100 runs. All ANN training was implemented using PyTorch [69] in Python.

### C.2   Computational Resources Used

All analyses were performed on one of two machines:

- **Machine 1:** Intel Xeon E5-2680 v4 @ 2.40GHz with 28 cores (56 virtual), and 256 GB RAM
- **Machine 2:** Intel Core i7-10700KF @ 3.80GHz with 8 cores (16 virtual), and 48 GB RAM

ANN training, information flow analysis, tradeoff analysis and scaling (dependency) analyses for the synthetic dataset (with kernel SVM for information flow estimation) all ran in approximately 11 hours and 34 minutes on Machine 2, when parallelized 8-fold (ANN training was not parallelized). The same four analyses for the modified Adult dataset on the smaller ANN (with linear SVM for

| Dataset | Machine # | MI estimator | Analyses | # ‖ᵉˡ Jobs | Time taken |
|---|---|---|---|---|---|
| Synthetic | 2 | Kernel SVM | ANN training | 1 | 11h 34m |
| | | | Info flow | 8 | |
| | | | Tradeoff | 8 | |
| | | | Dependency | 8 | |
| Modified Adult (smaller ANN) | 2 | Linear SVM | ANN training | 1 | 1h 25m |
| | | | Info flow | 8 | |
| | | | Tradeoff | 8 | |
| | | | Dependency | 8 | |
| Modified Adult (larger ANN) | 1 | Linear SVM | ANN training | 1 | 2h 49m |
| | | | Info flow | 10 | |
| | | | Tradeoff | 8 | |
| | | | Dependency | 10 | |

Table 1: Summary of analyses, computational resources and time consumed

Accuracy flow visualization
(Dataset: Synthetic, MI est: Kernel SVM)

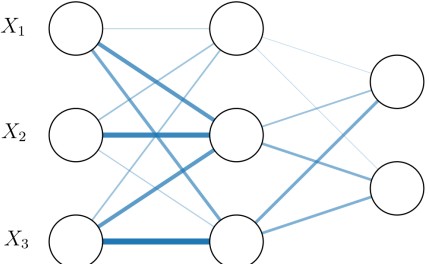

Bias flow visualization
(Dataset: Synthetic, MI est: Kernel SVM)

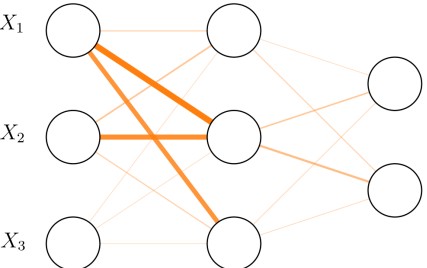

Figure 8: **Visualizations of accuracy and bias flows for the smaller ANN trained on the synthetic dataset.** Note how the most dominant accuracy flows arise from $X_3$, which is the only bias-free feature in the dataset. In contrast, the largest bias flows arise from $X_1$ and $X_2$, both of which are heavily biased features. It is intuitively clear from these pictures which edges have the largest bias-to-accuracy flow ratios, and hence which edges would be the first to be pruned.

information flow estimation) all ran in approximately 1 hour and 25 minutes on Machine 2, when parallelized 8-fold (ANN training was not parallelized). Finally, all four analyses ran for the modified Adult dataset on the larger ANN (with linear SVM for information flow estimation) in approximately 2 hours and 49 minutes on Machine 1; ANN training was not parallelized, information flow analysis and scaling (dependency) analysis was parallelized 10-fold, while the tradeoff analysis was parallelized 8-fold. This data is summarized in Table 1. In sharp contrast, the information flow and tradeoff analyses using the correlation-based estimate of mutual information ran in under 2 minutes in every case (time taken for training the networks is not included here).

## D   Additional Results

In this section, we provide several additional results extending the analyses presented in the main text. These are summarized below:

1. Figure 8 provides a visualization of the weighted accuracy and bias flows for the smaller ANN trained on the synthetic dataset. We focused on providing a visualization for the synthetic dataset, since we have some understanding of what the ground truth flows ought to be.

2. Figure 9 describes the dependence between weighted information flow on a given edge and the effect of pruning that edge on the output (i.e., a figure analogous to Figures 2 and 4), for the modified Adult dataset trained on the larger ANN.

3. Table 2 provides the Pearson correlation between the magnitude of information flow and the effect of intervening on an edge across all edges, for the dependency analyses shown in Figures 2, 4 and 9. The $p$-values for a statistically significant correlation are also shown, indicating that there is a consistent, statistically significant dependence between information flow and interventional effect for all but the last layer of the ANNs considered in these analyses. This consistency holds somewhat more poorly for the MNIST dataset, where the depth and width of the ANN result in only a very small dependence between pruning any individual edge and a change in the output accuracy or bias.

4. Analyses that span a more extensive set of pruning configurations described in Section 3.2, for the synthetic and modified Adult datasets (with the smaller and larger ANN). These are shown in Figures 10–12.

5. We performed tradeoff analyses where mutual information is estimated based on correlation, as described earlier in Appendix A.3. These are shown in Figures 13–15.

6. We also show a tradeoff plot with a randomized control: instead of removing nodes and edges on the basis of bias-to-accuracy flow ratios or accuracy-to-bias flow ratios, we select a fixed number of nodes or edges and remove them randomly. We do this for the modified Adult dataset on the smaller ANN for the correlation-based mutual information estimate. The result is shown in Figure 16, and matches with what we would expect: when nodes or edges are pruned randomly, the resultant tradeoff curve lies in between the curves corresponding to bias-to-accuracy flow ratios and accuracy-to-bias flow ratios. In particular, pruning on the basis of bias-to-accuracy flow ratio achieves a better bias-accuracy tradeoff than the randomized control.

7. We undertook an exploration of negative pruning levels, i.e., multiplying edge weights by negative numbers, on the modified Adult dataset for the larger ANN. The results of this experiment are shown in Figure 17.

A discussion on each of these results can be found in the respective figure captions.

**Remark on sequential pruning.** All of the pruning strategies presented in the paper pruning nodes, edges or paths in *parallel*, i.e., we reduced the weights of a number of selected edges or nodes *simultaneously* (where the number was determined based on the pruning level chosen). Since different pruning levels show very little difference overall, except in attaining lower accuracies or biases in the tradeoff plot, we did not present pruning strategies that pruned nodes or edges sequentially. The effect of sequential pruning can be intuitively extrapolated from the plots we present: for example, pruning two nodes sequentially should produce a tradeoff curve that exactly matches that for pruning the first node alone, following which, it would interpolate till it reaches the endpoint of the curve corresponding to pruning two nodes in parallel.

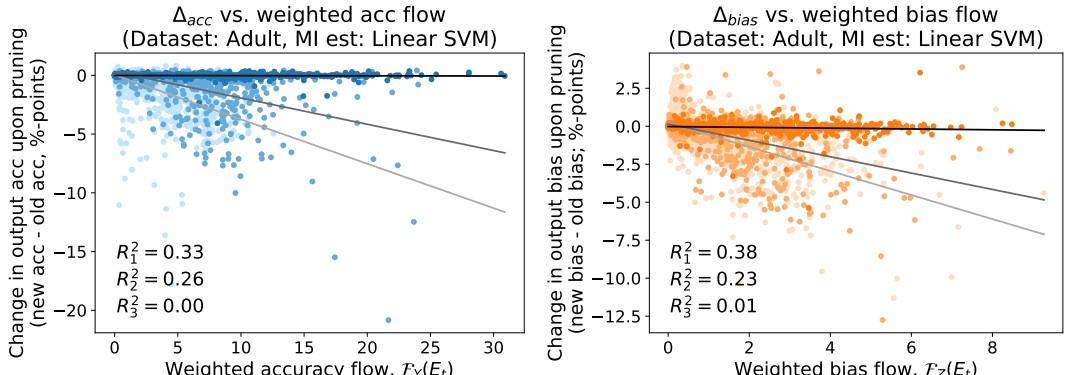

Figure 9: **Dependence plot for the modified Adult dataset trained on the larger ANN.** These figures show the relationship between the magnitude of information flow on a given edge, and the effect that pruning that edge has on the output of the ANN. Points in the lightest shade represent edges following the input layer while points in the darkest shade represent edges in the final layer of the network. We see trends very similar to those in Figures 2 and 4, where edges in the first two layers show a much stronger dependence than edges in the final layer. The lack of pruning effect for edges in the final layer might be attributable to the intrinsic redundancy in the one-hot encoding of the output layer, as mentioned in Section 4.2.

| Dataset | ANN | Flow type | Layer | Pearson correlation | $p$-value |
|---|---|---|---|---|---|
| Synthetic | Small ANN | Accuracy | 1 | $-0.35$ | $5.30 \times 10^{-27}$ |
| | | | 2 | $-0.20$ | $1.24 \times 10^{-6}$ |
| | | Bias | 1 | $-0.47$ | $1.52 \times 10^{-50}$ |
| | | | 2 | $-0.03$ | $4.18 \times 10^{-1}$ |
| Modified Adult | Small ANN | Accuracy | 1 | $-0.77$ | $1.81 \times 10^{-180}$ |
| | | | 2 | $-0.11$ | $6.89 \times 10^{-3}$ |
| | | Bias | 1 | $-0.52$ | $1.37 \times 10^{-63}$ |
| | | | 2 | $-0.18$ | $5.21 \times 10^{-6}$ |
| Modified Adult | Large ANN | Accuracy | 1 | $-0.57$ | $5.25 \times 10^{-132}$ |
| | | | 2 | $-0.51$ | $1.67 \times 10^{-61}$ |
| | | | 3 | $-0.04$ | $3.38 \times 10^{-1}$ |
| | | Bias | 1 | $-0.62$ | $2.54 \times 10^{-160}$ |
| | | | 2 | $-0.48$ | $6.30 \times 10^{-52}$ |
| | | | 3 | $-0.10$ | $1.00 \times 10^{-2}$ |
| MNIST | Deep ANN | Accuracy | 1 | $-0.12$ | $2.50 \times 10^{-13}$ |
| | | | 2 | $-0.19$ | $3.82 \times 10^{-29}$ |
| | | | 3 | $-0.11$ | $5.21 \times 10^{-12}$ |
| | | | 4 | $-0.02$ | $1.59 \times 10^{-1}$ |
| | | | 5 | $+0.03$ | $2.51 \times 10^{-1}$ |
| | | Bias | 1 | $-0.04$ | $8.65 \times 10^{-3}$ |
| | | | 2 | $-0.09$ | $1.78 \times 10^{-7}$ |
| | | | 3 | $-0.04$ | $1.47 \times 10^{-2}$ |
| | | | 4 | $+0.03$ | $5.12 \times 10^{-2}$ |
| | | | 5 | $+0.12$ | $1.63 \times 10^{-5}$ |

Table 2: Pearson correlation between information flow magnitude and interventional effect (i.e., correlation between $\mathcal{F}_Z(E_t)$ and $\Delta_{bias}$ upon pruning $E_t$, or correlation between $\mathcal{F}_Y(E_t)$ and $\Delta_{acc}$ upon pruning $E_t$, over all edges $E_t$), along with corresponding $p$-values, for the results in Figures 2, 4 and 9, and for the MNIST dataset. Correlations and $p$-values are computed using the Scipy `stats.pearsonr` function [70]; the $p$-value computation assumes Gaussianity, so $p$-values close to a desired significance threshold should be interpreted with some care.

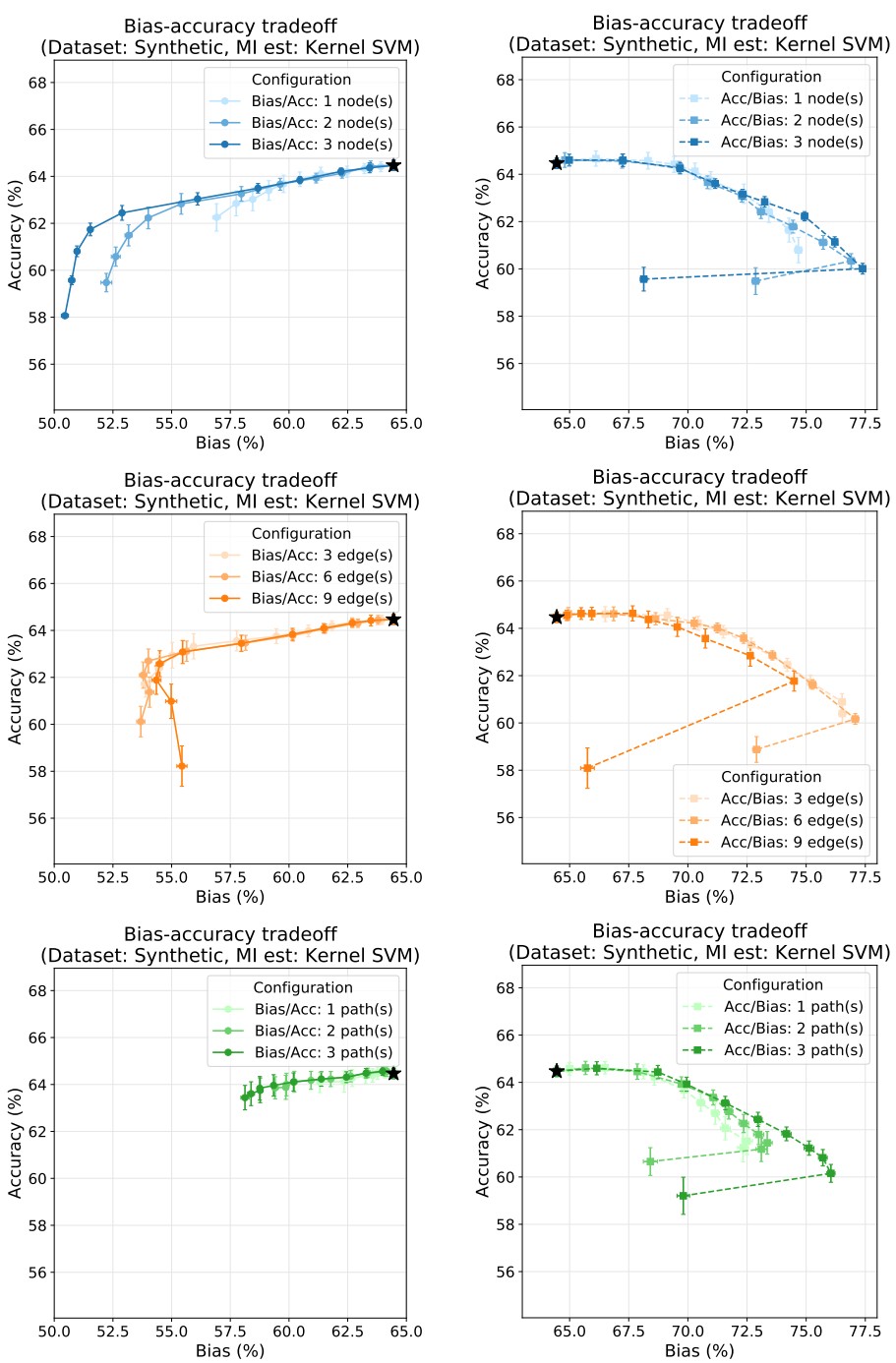

Figure 10: **Tradeoff plots for the synthetic dataset, with more pruning configurations.** These plots are very much in line with expectations, and are largely consistent with Figure 3. It should be evident that pruning nodes generally appears to outperform pruning edges. There appears to be a very limited difference for different pruning *levels* (refer Section 3.2), except that pruning more edges or nodes usually gives rise to lower accuracy and bias numbers. Occasionally, pruning at a higher level can also produce an overall better tradeoff plot, as in the case of pruning 3 nodes. Interestingly, pruning different numbers paths appears not to have a significant difference, even on the extent of bias or accuracy decrease, suggesting that the top three paths have a lot of overlap in their edge sets.

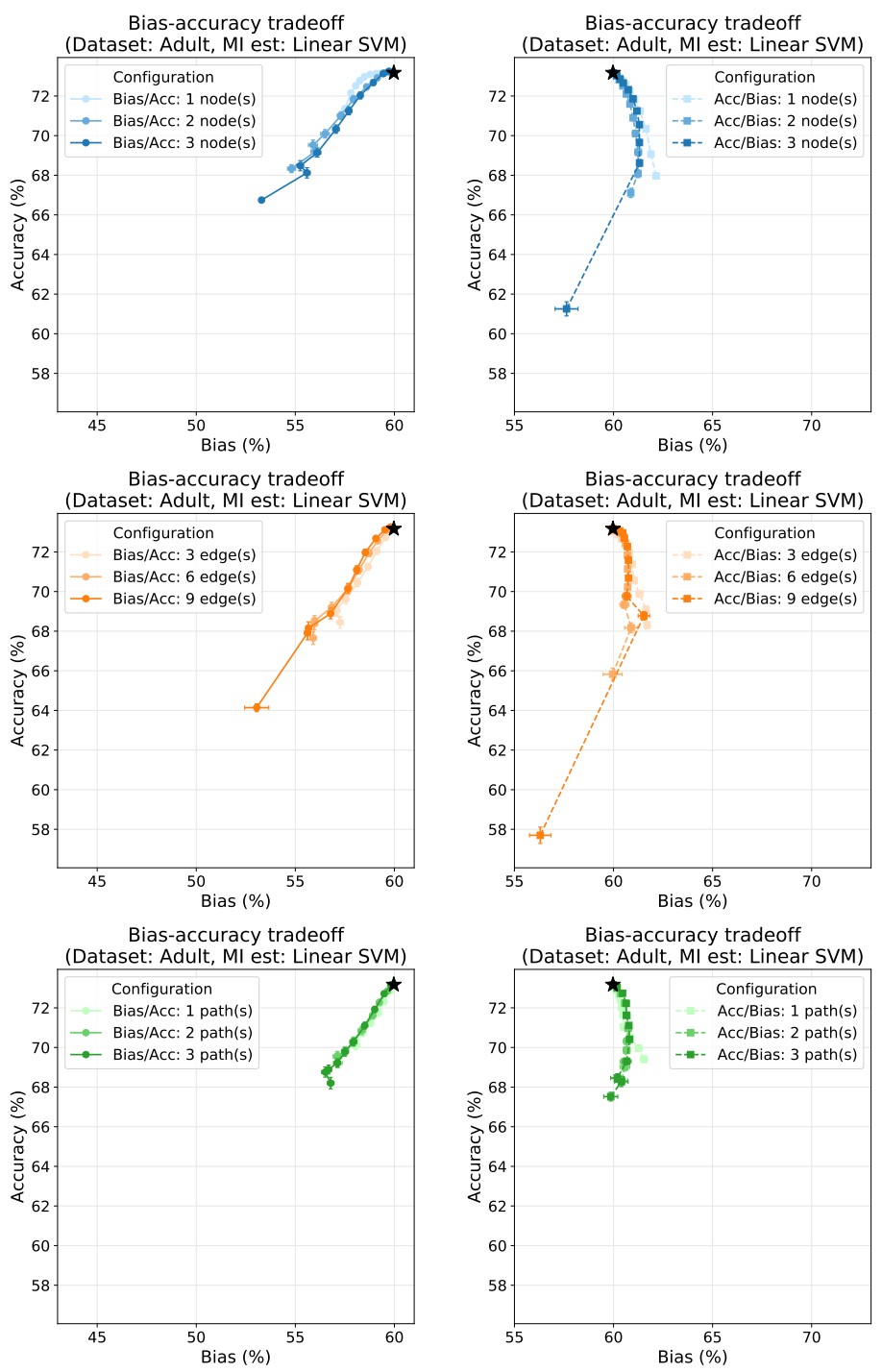

Figure 11: **Tradeoff plots for the modified Adult dataset trained on the smaller ANN, with more pruning configurations.** These plots are very consistent with those in Figure 5. Many of the observations mentioned in the caption of Figure 10 also carry over to this dataset.

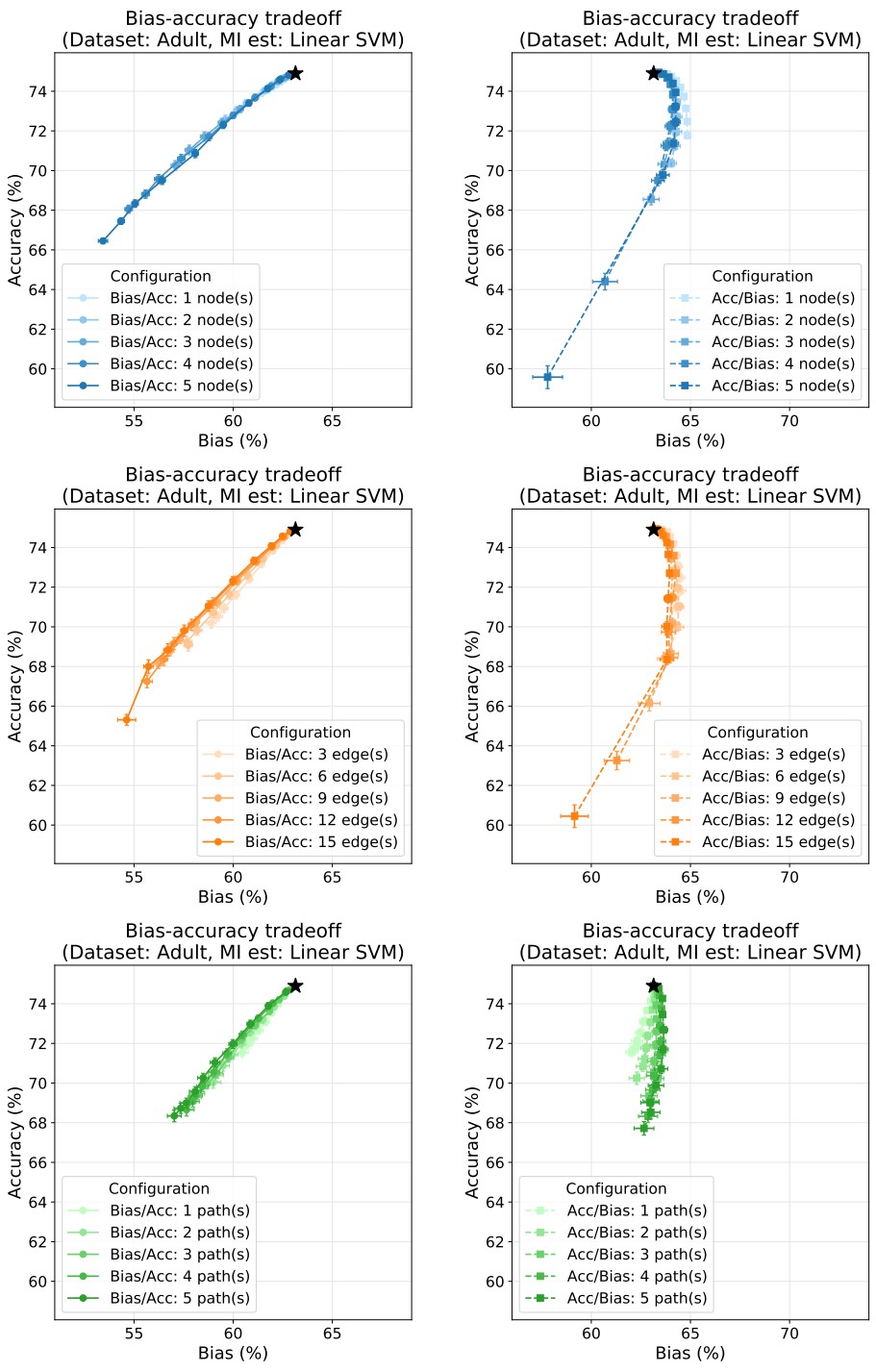

Figure 12: **Tradeoff plots for the modified Adult dataset trained on the larger ANN, with more pruning configurations.** These plots are very consistent with those in Figure 6a. Many of the observations mentioned in the caption of Figure 10 also carry over to this dataset.

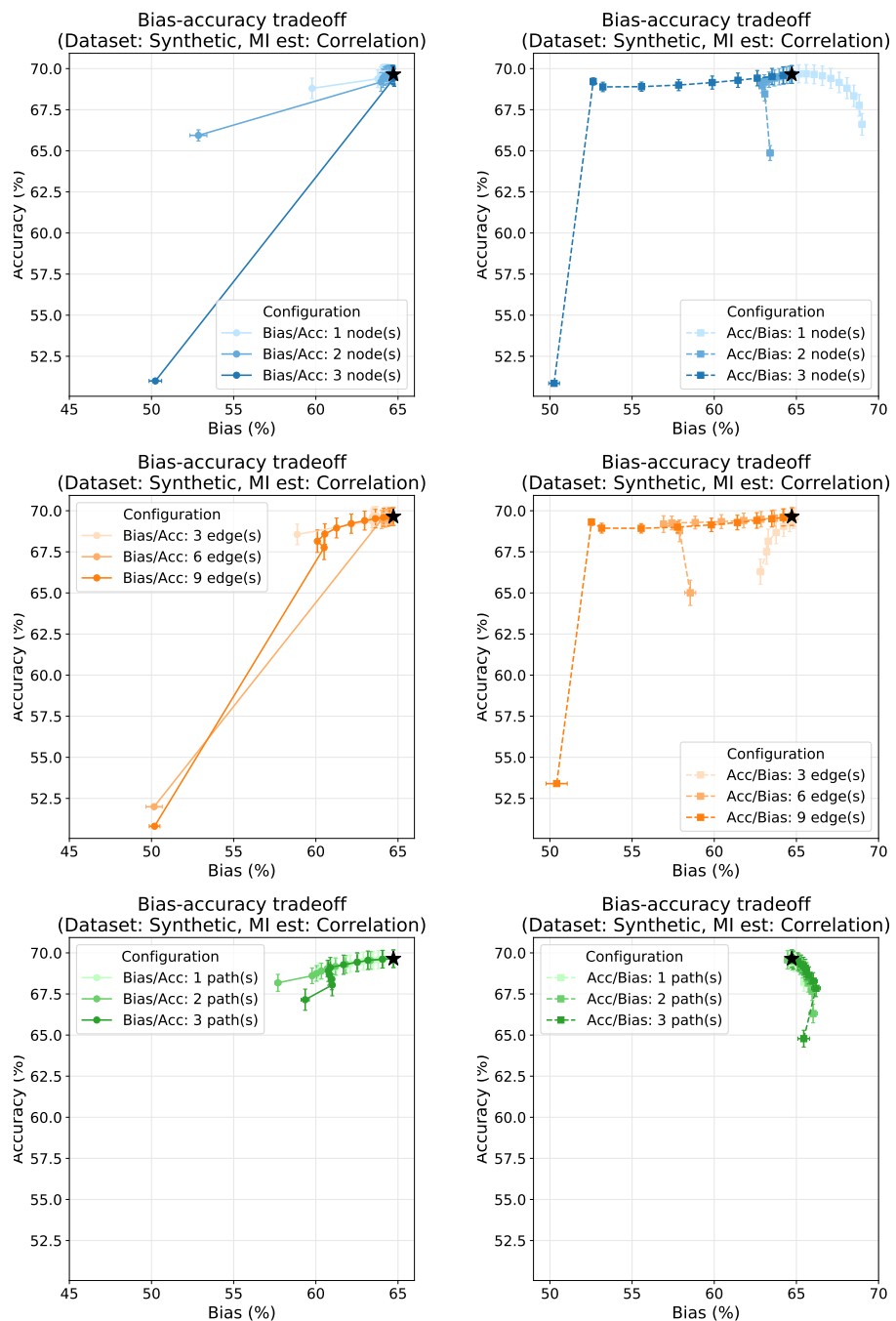

Figure 13: **Tradeoff plots for the synthetic dataset, with correlation-based information flow estimation.** These plots are *not* consistent with those presented in Figure 3 in the main paper. The erratic behavior that we see in several of the figures is a result of inverting poorly conditioned matrices, leading to large computational errors. As such, we present these plots to demonstrate a failure case for correlation-based information flow estimates. When the dataset has an intrinsic non-linearity (as described in Appendix B.1), correlation-based estimates of mutual information are very poor. The linear-SVM based classifier also failed on this dataset for the same reason (as mentioned in Appendix A.1).

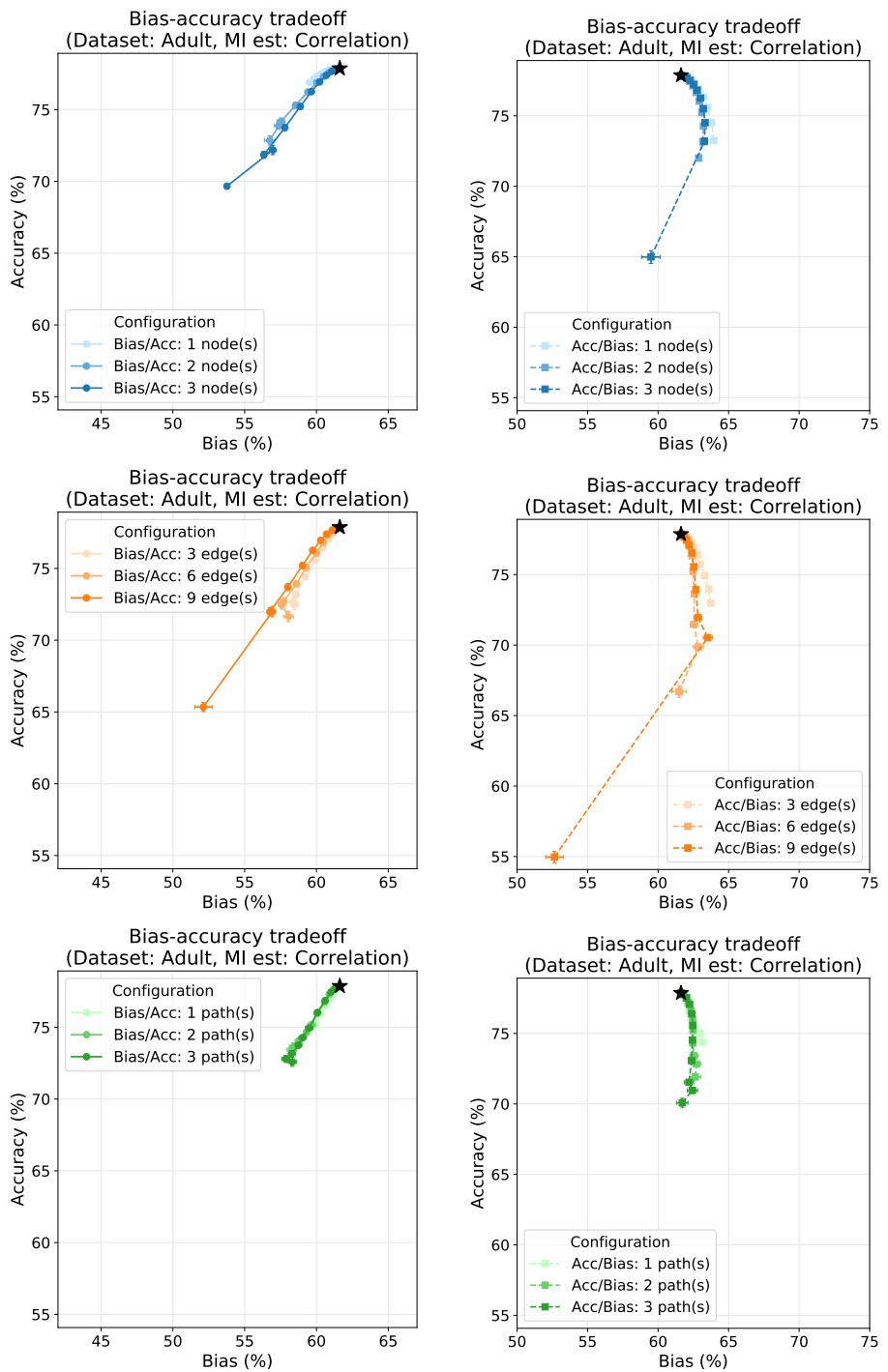

Figure 14: **Tradeoff plots for the modified Adult dataset trained on the smaller ANN, with correlation-based information flow estimation.** These tradeoff plots are consistent with those in Figures 5 and 11. This suggests that most of the dependence in the modified Adult dataset can be described through linear relationships, which is also why linear SVM was our classifier of choice. As an important difference between the correlation-based estimate in this figure and the linear-SVM–based estimate in Figure 11, note that the correlation-based technique *overestimates* the mutual information. However, the *qualitative* aspects of the tradeoff curves are preserved, including the differences between different pruning levels. This suggests that the correlation-based estimator is useful when we want a qualitative understanding of how the tradeoff would behave, while allowing for obtaining results in a fraction of the total time taken by the more compute-intensive classifier-based methods.

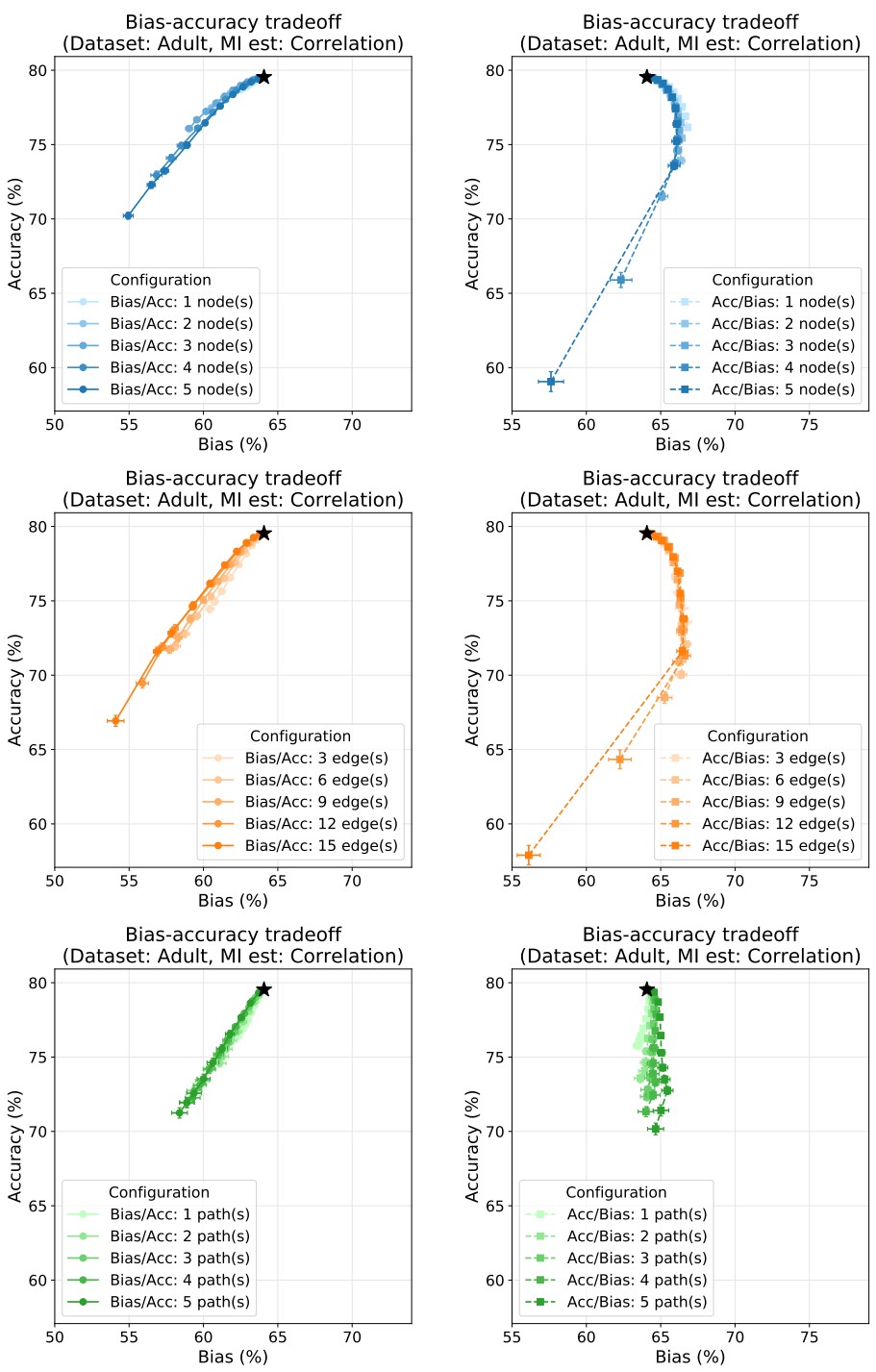

Figure 15: **Tradeoff plots for the modified Adult dataset trained on the larger ANN, with correlation-based information flow estimation.** These tradeoff plots are consistent with those in Figures 6a and 12. Once again, the correlation-based estimate overestimates mutual information, while capturing the qualitative features of the tradeoff plots. The conclusions from Figure 14 for the smaller ANN carry over to the larger ANN shown here as well.

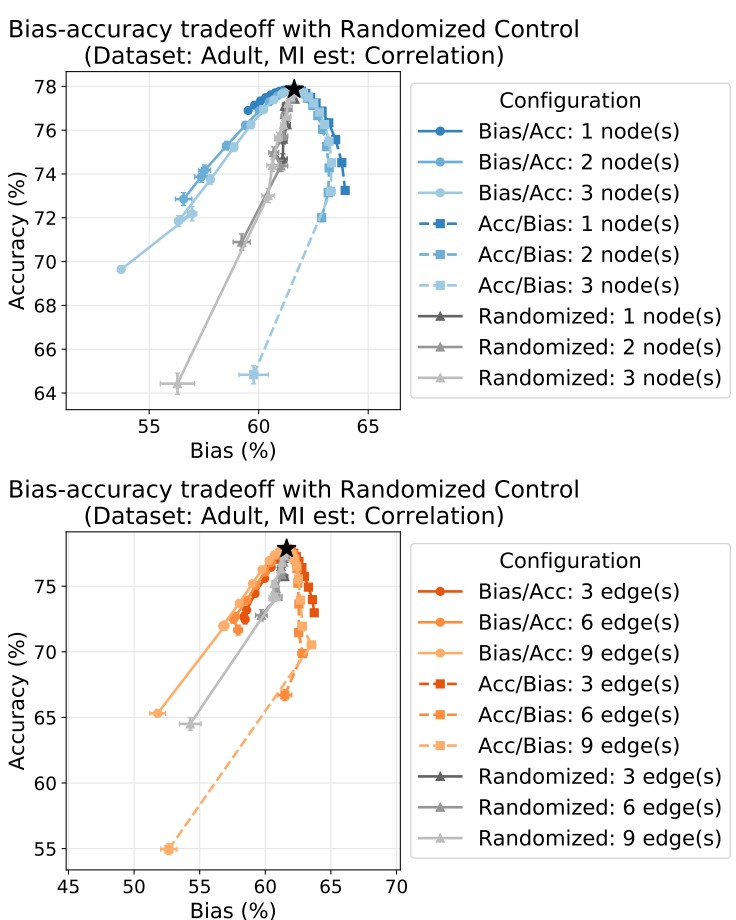

Figure 16: **Tradeoff plot for the modified Adult dataset trained on the smaller ANN, using correlation-based mutual information estimation, against a randomized control.** To provide an additional control condition and examine whether pruning on the basis of bias-to-accuracy flow ratios achieves a better tradeoff than pruning nodes or edges randomly, we implemented a randomized control. We selected the same number of nodes (top) or edges (bottom) to be pruned on the basis of largest bias-to-accuracy flow ratios, largest accuracy-to-bias flow ratios, and chosen at random. The results match what we would intuitively expect: the tradeoff curves for the randomized control condition fall in between those corresponding to bias-to-accuracy flow ratios and accuracy-to-bias flow ratios. Further, pruning on the basis of bias-to-accuracy flow ratio achieves a better bias-accuracy tradeoff than pruning the same number of nodes or edges randomly.

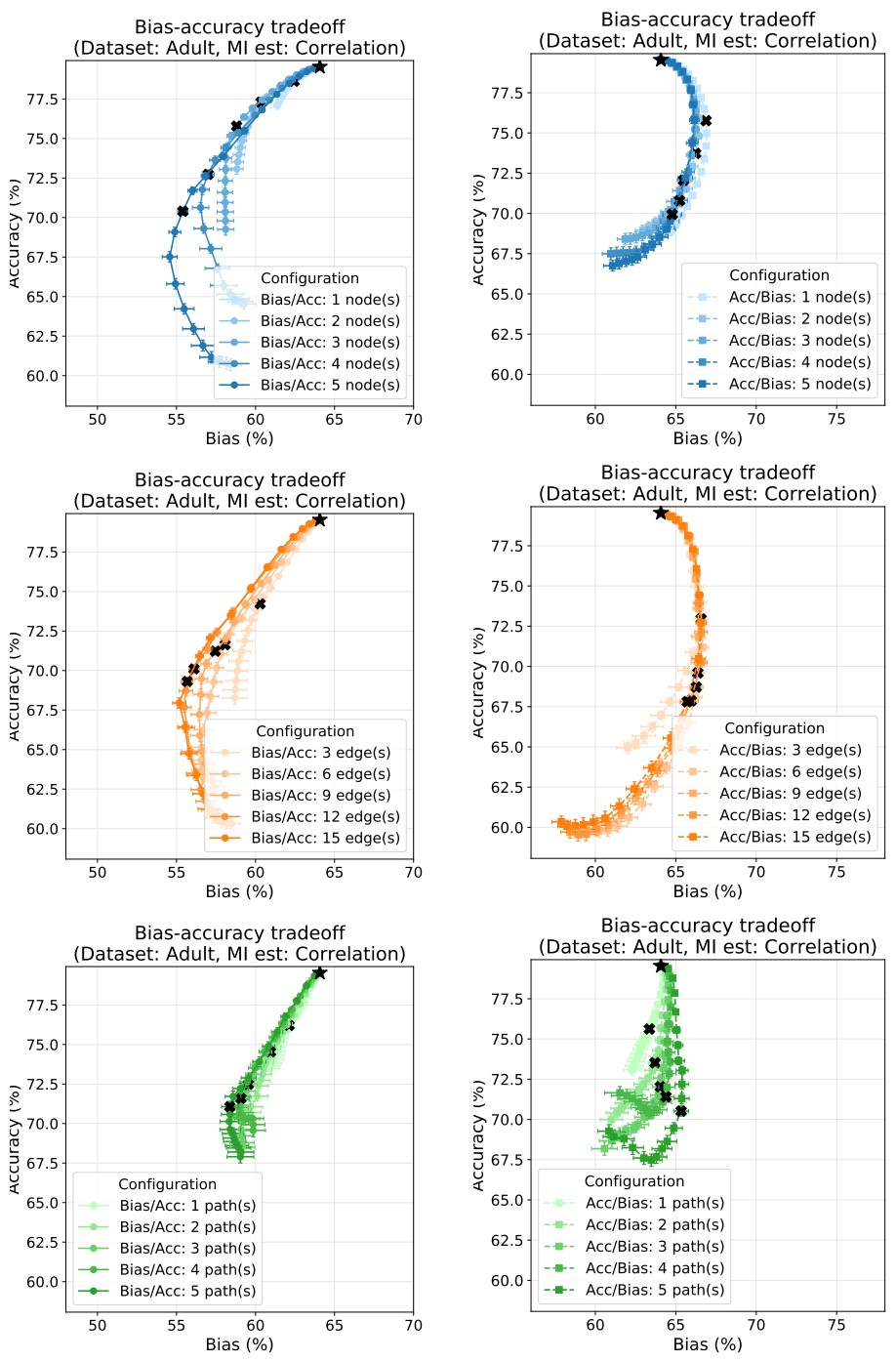

Figure 17: **Tradeoff plots for the modified Adult dataset trained on the larger ANN, using correlation-based mutual information estimation, and with negative pruning levels.** To examine the impact of "pruning below zero", i.e., multiplying edge weights by negative factors, we considered the correlation-based estimate, since this suffices to obtain a qualitative understanding. The ✖ denotes the point at which the edge is completely pruned, following which edge weights are multiplied by increasingly negative fractions: $[-0.1, -0.2, \ldots, -1]$. Note how negative pruning factors produce undesirable effects: in the case of bias-to-accuracy flow ratios (left column), negative factors start *increasing* the bias once again.