# OpenReview forum: "Can Information Flows Suggest Targets for Interventions in Neural Circuits?"
_NeurIPS.cc/2021/Conference — NeurIPS 2021 Poster_

### Official Review · Reviewer_GutV · 2021-07-15

**Rating:** 6
**Confidence:** 3

**Summary:**

The paper studies the possibility of using the measures of information flows to intervene with ANN for reducing bias while improving/maintaining accuracy. It adopts the M-information flow framework proposed by Venkaatesh et al., and proposes a quantitative notion of information flow and a method for estimating such flows. The magnitude of information flows can then be used to understand and compare the impact of different pruning strategies on the outputs, as well as their fairness-accuracy tradeoffs.

**Limitations And Societal Impact:**

Yes

**Main Review:**

Overall the paper is well-written and easy to understand. The methods are simple and easy to use.

My main concerns are as follows.

1. Most techniques are adopted from the existing works such as the concept of M-information flow, method of estimating information flow based on Fano’s inequality, and typical pruning methods, etc. One major concern I have is the novelty of the paper.
2. Although the methods seem to be effective on synthetic data and modified adult data, there is no theoretical support/guarantee of the method. Moreover, the experiments only consider relatively small networks (3 features, one hidden layer & 5 features, 2 hidden layers). It is not clear how this method performs on other datasets and larger networks, whether the improvement and effectiveness still preserve in more general settings.
3. It seems that the method can only be used for the notion of demographic parity fairness, which may limit the generality of the proposed model.

Additional questions:

1. Information flows are quantified by taking maximum over all subsets of nodes in a layer (Eqn (3)). Are there any particular reasons for taking maximum rather than other operations such as average?
2. The experiment on synthetic data only considers a small neural network (3 features, one hidden layer). Figure 3 shows that there is no significant difference between different pruning methods regarding the fairness-accuracy tradeoff. Is it possible it’s because the network is too small? I suggest authors performing experiments on a larger-scale network and compare accuracy/fairness between different methods. Intuitively, why does fairness-accuracy tradeoff show the contrarian patterns using pruning on the basis of accuracy-to-bias flow ratio and pruning on the basis of bias-to-accuracy flow ratio?



**Time Spent Reviewing:**

6

---

> ### Author Response · Authors · 2021-08-11
> **Responses to Reviewer GutV**
>
> We thank the reviewer for their careful review of our paper.
>
> **Novelty**
>
> The novelty in our paper is two-fold:
>
> 1. First, the M-information flow framework of Venkatesh et al. is purely theoretical, and to our knowledge, there has been no concrete validation/use of their work in either simulated or neuroscientific settings. Our paper takes the first step towards a practical application of their theory, by showing how information flows can be estimated in an artificial neural network on synthetic and real datasets. That the framework allows tracking information flow of differences messages is directly useful in the use case as well as examples discussed here.
> 2. The second contribution has to do with relating information flow (an observational quantity) and interventions: we show that the magnitude of M-information flow on an edge is correlated with the extent of reduction in bias (or accuracy) upon pruning that edge. This question was not investigated by Venkatesh et al. In fact, it is not possible to _prove_ such a link between information flow and interventions: there are always counterexamples where pruning an edge with a large information flow could have no effect on the output. However, our work shows empirically that, _in practice_, information flows do appear to predict the effect of interventions.
>
> **Theoretical guarantees**
>
> The theoretical guarantee for information flow itself (existence of information paths) was provided by Venkatesh et al. It is not possible to provide a similar guarantee linking information flow and interventional effect (see previous point). We will also add a counterexample to illustrate this in the supplementary material, for the final version of the paper.
>
> (We will update this comment with the exact wording of the counterexample in the next 24 hours).
>
> **Extending to larger networks**
>
> The current work is intended to be a proof-of-concept showing the use of M-information flow, and applying it to determine where to intervene in a decision-making network. We focused on small networks for a few reasons:
>
> 1. Smaller can be more fully analyzed: we can get a better intuition about what the network is doing, so we can check if M-information flow is working in a reasonable manner.
> 2. Many "canonical" computational circuits in neuroscience are actually quite small: they consist of a small number of brain areas interacting with each other to perform a particular task. One example of this might be the well-known reward network, which consists of 11 nodes with 17 edges between them (see Fig.1 in Russo and Nestler, Nat. Rev. Neurosci. 2013). In such examples, it suffices to intervene and disrupt the activity between a few of these regions in order to have an effect on how the brain processes information. Evidence for this exists in studies that have used targeted electrical stimulation to treat disorders of the reward network (e.g., in addiction [Luigjes et al., Molecular Psychiatry, 2012], depression [Delaloye et al., Dialogues Clin Neurosci. 2014], obsessive-compulsive disorders [Border et al., Ment Illn. 2018]).
> 3. Even in non-neuroscientific application domains, small decision-making networks are often used. For example, for the datasets we considered (the synthetic dataset, the adult dataset with three features and the adult dataset with five features), we tested to ensure that increasing the depth of the neural network did not improve accuracy. So, for similar applications, we expect that examining and understanding small networks is also of interest.
>
> Nevertheless, the reviewer's question is also reasonable and addressing larger networks has practical value:
> 1. Our primary motivation behind considering networks of two different sizes was to understand the effect of going to larger networks. One interesting observation we made, for instance, was that when analyzing a larger network, a larger number of interventions are needed to achieve a similar trade-off.
> 2. We also discuss how the _layer_ in which an edge lies affects the dependence between its information flow and interventional effect (see Lines 304-311; also see Figure 9 in the supplementary material to compare with the larger network).
> 3. An important component of scaling to larger networks is the amount of time taken to compute mutual information. In an effort to test whether and when simpler measures might suffice, we checked how well correlation is able to approximate mutual information (see Appendix A.3, Appendix D, and Figures 13-15 in the supplementary material). To summarize, correlation works reasonably well when there are no severe non-linearities in the dataset, as we might expect. As a result, correlation fails in the case of the synthetic dataset, where we purposefully introduced a nonlinearity, however it works well in the case of the modified Adult dataset.
> 4. We will also try to include an analysis of a deeper network (~5 layers) in the final version of the paper. If we are able, we will update the reviewers with more information about this during the rolling discussion section.
>
> **Other fairness measures**
>
> As stated in our response to Reviewer BPew, we believe that our method can probably be extended to include some other definitions of fairness which can be represented in terms of mutual information. For example, we believe our measure can be extended to equalized odds as well, by using an appropriate modification of the M-information flow definition. The interpretation of such a quantity as an "information flow" might be called into question, however, a suitable tradeoff between equalized odds and accuracy might still be found.
>
> **Responses to additional questions**
>
> 1. The intuition behind taking the maximum over all subsets comes from the idea of "synergy". Essentially, an edge $E_t$ is said to have M-information flow either if $X(E_t)$ has mutual information with $M$ by itself, or if $X(E_t)$ _along with_ some other edge $X(E_t')$ has mutual information with $M$. This latter case is called a _synergistic interaction_ between $X(E_t)$ and $X(E_t')$. Taking a maximum over all subsets is equivalent to considering the maximal synergistic contribution from all other subsets of edges.
>
>     A better way to measure this might be to directly estimate synergistic information and add up all contributions, however there is still much debate between different definitions of synergy, and the efficient estimation of synergistic information is still a very nascent field.
>
> 2. (a) The comment about larger networks has been addressed above. Also see our response to Reviewer YNbt.
>
>     (b) This is a great observation. As we noted briefly in Section 4, the reason we see the contrarian patterns is because the network was trained for _accuracy_. Therefore, assuming we obtained the maximum possible accuracy, all of the $Y$-information present in features was already being extracted to produce the output $\hat{Y}$. However, the dependence between $\hat{Y}$ and $Z$ was not maximized, and a potentially greater dependence with $Z$ was possible by using a different set of feature weights. By intervening on edges with larger accuracy flows, we decreased the influence of $Y$; for the datasets we considered, this also happened to mean that the dependence on $Z$ automatically increased.
>
>     If the network had been originally trained for bias, we would likely have seen the same contrarian pattern, but with accuracy and bias flipped: pruning edges with maximal bias flows would have resulted in an increase in accuracy. We explained this briefly (e.g., see lines 278-281), however, in the revised version, we will further emphasize this.

---

> > ### Comment · Reviewer_GutV · 2021-08-26
> > **Response to authors**
> >
> > First, I want to say that I understand and acknowledge the authors' response, and I am sorry for the delay in replying to them.
> >
> > Regarding the lack of theoretical analysis and guarantee, I appreciate the additional explanations and counterexamples provided by the authors and I agree it could be challenging to quantify the relations between information flow and interventions theoretically. I also appreciate the additional experiments on larger networks. I am willing to increase my score to weak acceptance.

---

> ### Author Response · Authors · 2021-08-12
> **Counterexamples in lieu of theoretical guarantees**
>
> Below, we provide two counterexamples to show the challenges associated with providing theoretical guarantees linking information flow and interventions. As such, these counterexamples illustrate why we took an empirical approach in this paper.
>
> > The first counterexample is intended to show that the presence of information flow on a particular edge, no matter how large, does not by itself imply that pruning that edge will produce an effect at the output of the network.
> >
> > Consider an artificial neural network that generates an intermediate feature which is strongly correlated with the protected attribute, and thus has a large information flow. Suppose also that this feature does not contribute to $\hat{Y}$ (e.g., because all weights leading out of this node are zero). In this case, pruning this node will naturally have no effect on the bias at the output, even though it can have an arbitrarily large information flow.
> >
> > &nbsp;
> >
> > Next, one might be interested in asking whether the converse of the above statement is true instead: is it possible that if a particular edge has no information flow about a message, then pruning that edge will have no effect upon the dependence of the output on the message? Unfortunately, this statement is also not true.
> >
> > Consider an artificial neural network where the message of interest, $M$, and another independent random variable $Z$ (to be precise, assume $Z$ is not causally influenced by $M$) arrive at the same neuron (say with ReLU activation) somewhere in the middle of the network. Then, provided the weights of both edges is comparable, we can use $Z$ to "gate" whether or not $M$ flows through the network. For instance, if $Z$ is 0, then the neuron in question will activate based on $M$ alone. However if $Z$ is an extremely large negative number, then the neuron is unlikely to ever be activated if the distribution of $M$ lies in a more reasonable range. Thus, pruning the edge corresponding to $Z$ can determine whether information about $M$ is passed on to the output or not, even though $Z$ itself has no information flow about $M$.
> >
> > &nbsp;
> >
> > Note that the above counterexamples are theoretical. In particular, we do not make assumptions about whether or not a network may actually attain such a state, where a particular feature is _generated_ but not _used_ later on, or where a "gating" variable is in play. The reason we make no such assumptions is that, in both the neuroscientific and AI contexts, we do not fully understand what the brain or the ANN might do. Especially in cases where the brain or AI algorithm is _faulty_, unexpected flows may arise.
>
> These counterexamples illustrate why it is hard to provide any clear theoretical guarantees about how information flow is related to interventions. This is why it is essential that our paper studies this question _empirically_ instead.

---

### Official Review · Reviewer_BPew · 2021-07-16

**Rating:** 7
**Confidence:** 3

**Summary:**

This paper studies whether observational data can be used to estimate the outcomes of certain interventions. This is especially important since cost of interventions can be high and in some cases irreversible. This paper empirically demonstrates the effectiveness of using the M-information flows framework for neural networks to assess different kinds of information flows in the network, and then measuring the change in outcome by intervening on different types of flows. They consider fairness in ML as a case study where there are two types of distinct information flows: the label flow and the bias flow. By intervening on paths (by pruning nodes or edges or reducing weights of edges) which contribute to bias in the outcome, the authors show that they can achieve the desired effect of reducing bias. Thus, the paper shows that information flows can offer useful information about where to intervene to achieve certain outcomes.

**Limitations And Societal Impact:**

The authors have acknowledged the limitations and potential societal impacts.

**Main Review:**

This paper addresses a very important problem that comes up in all kinds of application areas (fairness, healthcare, drug discovery etc). This  paper empirically demonstrates how to use information flows in finding neurons/edges which can be intervened upon to produce a desired outcome. I like using fairness as a use case for such a framework since there are two distinct kinds of flows that can be measured using well defined metrics. This also offers a new perspective on the role of model's structure in contributing to biased outcomes. The methods used in this paper are broadly applicable to other areas of interest in ML and Neuroscience. The paper is also very well written and provides all necessary details, and is thus accessible to a broad audience. Overall, I think this is a very interesting paper.

### Scalability

 * All the neural nets used in experiments are quite toy-ish, with very few layers and very few nodes. I understand that this allows the authors to extensively evaluate all kinds of flows, however, how would this scale to larger, deeper nets with number of paramaters that are beyond the scope of manual inspection?

 * How would size of NNs affect the tradeoffs presented in Figures 3 and 5?

 * In reality, even for the case of fairness, one would want to consider intersectional attributes. This would be true even in domains such as drug discovery. How would the proposed method scale for non-binary outputs and non-binary protected features?

### Other

 * The paper only considers demographic parity (which is fine), however can this be adopted for other definitions like equal odds and equal opportunity?

 * I know this isn't the goal of your paper, but what you're doing by intervening to on Z-information flows sounds similar to some of the adversarial learning methods to remove the effects of the sensitive attribute on the output (eg: https://arxiv.org/abs/1807.00199; https://arxiv.org/abs/1902.00334). It would be interesting to see the effectiveness of pruning (your method) vs adversarial methods.

**Time Spent Reviewing:**

4

---

> ### Author Response · Authors · 2021-08-10
> **Responses to Reviewer BPew**
>
> We thank the reviewer for their comments. Responses to individual points raised follow:
>
> **Scalability**
>
> To address the reviewer's questions, we will add a subsection on scalability to our discussion section in the final version of the paper.
>
> 1. As the reviewer points out, we chose small networks for this proof-of-concept paper precisely because it is possible to gain an intuition for what is happening within the network, and check whether the information flows we see make sense.
>
>     In the paper, we make an effort to discuss the effect of increasing the breadth and depth of the network by considering the slightly larger ANN with two hidden layers. The computational cost for larger networks is also very high; to address this, we have discussed whether mutual information can be approximated using a correlation-based estimate, and to what extent this works - these results can be found in the supplementary material.
>
>     We will also try to include an analysis of a deeper network in the final version of the paper, and update the reviewers with our results in the rolling discussion period.
>
> 2. Regarding the tradeoff curve, we noticed an interesting point between our smaller and larger ANNs: with increasing size, the number of interventions required (i.e., number of nodes or edges we need to prune) to achieve a reasonable tradeoff also increases. This effect is likely to scale for larger networks. We will incorporate this observation in our discussion section.
>
> 3. For considering flows where M is non-binary (e.g., intersectional attributes), our core methodology does not change. However, different methods may need to be used for estimating mutual information, suitably tailored to the type of dataset involved: there is a broad literature on this subject (e.g., Kraskov et al., Phys Rev E 2004; Gao et al., arxiv:1709.06212, 2018; Belghazi et al., ICML 2018; Mukherjee et al., UAI 2020; Mondal et al., UAI 2020; etc.). Another important issue that arises with intersectional attributes in real datasets is that sample size may start becoming very small. This may result in poor information estimates, so the variable M should be chosen with care.
>
> **Other**
>
> 1. Since we were mainly thinking of implications to neuroscience, we only considered M-information flow as defined by Venkatesh et al. The reason the fairness context works so well is that the standard definition of information flow has a natural interpretation as a fairness measure, namely demographic parity.
>
>     To cater to other fairness measures, we have to consider how to modify the original M-information flow definition. For example, equalized odds has been interpreted in terms of conditional mutual information, e.g., $I(M ; \hat{Y} | Y)$ where $M$ is race, $\hat{Y}$ is the output and $Y$ is the true label [cite]. Similar conditioning on $Y$ for measuring the "equalized-odds-bias-flow" would probably work (e.g., $I(M ; X(V_t) | X(\mathcal{V}_t'), Y)$); however, it is not clear if we can truly interpret this as a quantity representing "information flow". Regardless, it might work as a heuristic for measuring the tradeoff. We will add a short discussion on this to the paper.
>
> 2. We thank the reviewer for providing these references; we will cite both of these works and discuss briefly. Indeed, these works are similar to ours in spirit: however any method that uses information about bias to directly penalize and re-train the original network will probably do better than our method, which does not re-train. Nevertheless, understanding the effects of re-training may also have implications for neuroscience, where interventions may not have lasting effects if the brain is able to re-learn undesirable associations. These are interesting directions to consider for future work.

---

### Official Review · Reviewer_YNbt · 2021-07-21

**Rating:** 6
**Confidence:** 3

**Summary:**


This paper adapts the concept of information flows, develops by Venkatesh et al for general computational systems, to (artificial) neural networks with the goal of studying how to "intervene" in an already trained NN to change the way its output is affected by different inputs. The authors then apply this approach, namely a pruning-based approach for altering information flows, in order to manage the bias-accuracy tradeoff in fair classifiers that take sensitive features as inputs.

**Limitations And Societal Impact:**

Limitations and social impact are adequately addressed in the discussion section.

**Main Review:**


Overall, the paper presents a very interesting approach to manipulating the label-feature interactions for the sake of tuning the fairness-accuracy tradeoff in neural nets. There are some interesting contributions in the paper, namely the formulation of an information-theoretic counterpart of (demographic) parity, and the development of a simple pruning-based algorithm for evaluating the bias-accuracy tradeoff. On the flip side, the following drawbacks makes me reluctant to fully recommend the paper for acceptance. These issues are listed below:

**+ Confusing introduction, focus and an unclear motivation/objective**

The paper's title, abstract and introduction do not fully reflect what the paper is about. In my view, the key contribution of the paper is proposing an information-theoretic approach to evaluating/ensuring fairness of a neural net **classifier** for a given level of accuracy. The abstract and introduction discuss "interventions" from observational data---in the clinical and neuroscientific context---in a way that suggests learning the causal pathways of clinical interventions (Lines 4-7, Lines 21-24). But the concept of an intervention here is very different as it refers to alterations to the network architecture rather than manipulation of actual interventions in the real world.

Also, the paper presents fairness as a potential application of the broader methodology of information flows. However, it seems to me that the proposed method has very limited applications beyond that, and it only works for simple binary classifier as estimating the conditional mutual information $I(Y|Z)$ is practically impossible for more complex tasks. I think it would have been better if the paper was written with a focus on the "fairness of classification" application.

**+ Experiments: absence of baselines**

The authors used the experiments section to show that information flows can be used to find targets for intervention in the context of evaluating the bias-accuracy tradeoff. However, I think a more appropriate research question is whether this is the right way, or a better way, to evaluate this tradeoff. There are many methods for evaluating and ensuring fairness in classifiers that have been proposed in the past few years, that also have the advantage of being model-agnostic and not restricted to neural nets. I think the authors should have compared with these.

Minor issue: I am also unsure about the validity of the linear fits in Figure 4 given the massive variance and the lack of a clear trend.

















**Time Spent Reviewing:**

5+ hours

---

> ### Author Response · Authors · 2021-08-11
> **Responses to Reviewer YNbt**
>
> We thank the reviewer for their time and their thoughtful review of our paper.
>
> **Clarity in title, abstract and intro:**
>
> We thank the reviewer for raising this issue. We will clarify the text to explain the motivation and the goal better, as substantiated under the following subheadings. (We will also follow up in the next 24 hours with the newer wording that we intend to use in the introduction.)
>
> **Key contribution - fairness vs. neuroscience:**
>
> The reviewer's view of the key contribution is well taken, however, we believe that the value of the paper lies elsewhere for a few important reasons.
>
> We believe the value of our paper lies in showing that observational measures can allow one to predict where interventions might need to be made:
> 1. Venkatesh et al. provided a purely theoretical framework for studying information flow. The first contribution of our work lies in validating this theory through proof-of-concept experiments on Artificial Neural Nets with synthetic and real datasets, which is the next step in making this theory practically applicable.
> 2. The second, and perhaps more important, contribution of our paper lies in showing that information flow can actually tell you where to intervene in a network to change its output.
>
> While the proposed method might eventually also find use in tuning bias-accuracy tradeoffs, we believe that there are more direct techniques that allow for that, e.g., incorporating measures of fairness into the objective function while training a neural network. Such methods, which address the fairness-vs-accuracy problem more directly, will probably outperform an indirect method such as ours: both in the actual tradeoff achieved, as well as in the amount of time required to produce a network that achieves such a tradeoff. A serious application of our work in the fairness domain will require substantial research effort comparing our pruning technique with these works, which is beyond the scope of the current paper.
>
> The overarching goal of our paper is: how do you minimally edit a decision-making network towards some objective? This is also why we don't compare with other fairness methods: in all likelihood, those would do better at the fairness task, but they would not tell you how to edit the network.
>
> **Interventions in neuroscience**
>
> Contrary to what the reviewer suggests, modern interventions in neuroscience increasingly and very closely resemble alterations to network architecture, e.g. specific nodes or edges can be activated/deactivated using optogenetics. Minutely invasive techniques, such as neural dust, are being tested in animal studies. Minute probes are already used invasively for intervening on the reward network (e.g., in addiction [Luigjes et al., Molecular Psychiatry, 2012], depression [Delaloye et al., Dialogues Clin Neurosci. 2014], obsessive-compulsive disorders [Border et al., Ment Illn. 2018]) and movement control network (in Parkinson’s [Vingerhoets et al. Neurology’02]).
>
> When we say "interventions in clinical settings", we are referring to interventions to treat neurological conditions, which attempt to change network structure by altering what a node computes or communicates (e.g., disruption of feedback cycles to treat epilepsy/depression/etc.). In particular, we do not mean "interventions" in the sense of "treatments" (e.g., through diffuse medication) that is sometimes used in the classical Causality literature. That body of literature is usually not concerned with connecting observational measures with interventional effect. Since our work is also closely related to Causality, we will clarify so as to avoid ambiguity between these interpretations.
>
> **Broader applications:**
>
> As mentioned in the paper, we believe that the strongest connection of this paper is with neuroscientific applications, where there is a great need for measures that can be used to observe neural activity, and then allow one to predict where changes should be made within a network, so as to change this activity. There is also a need to distinguish between _desirable_ and _undesirable_ flows of information, and edit the network to preferentially affect one over the other. The fairness example is an excellent analogy to this problem, which is why we use it as a case study in our paper. However, we do not mean to restrict ourselves to it.
>
> (We will also follow up in the next 24 hours with an example that more clearly brings out the connection between these fields, and clarifies why the fairness context is such a good fit for the neuroscientific setting. This example will be used at the very start of the introduction to motivate the paper, and will hopefully resolve the reviewer's concerns with clarity in the introduction.)
>
> **Baselines:**
>
> As mentioned above, we didn't compare with other fairness methods because we felt that the question that we were addressing was very different, i.e., the question of _how to edit the network_ (keeping the neuroscientific/clinical end goal in mind), rather than how to induce fairness in a decision making system.
>
> However, in lieu of a baseline, we will also compare with the effect of randomly selecting edges to prune (as a kind of randomized control, to evaluate the extent to which the knowledge of flows is actually helping), and add these results to the supplementary material in the final version of the paper. Indeed, the results for pruning based on accuracy-to-bias flows already perform this role to a degree, and one might imagine that a randomized method will pick some edges from each set, producing tradeoff curves somewhere in the middle of the two.
>
> **Linear fit:**
>
> The purpose of the linear fit was to clearly illustrate the presence of dependence: the magnitude of information flow on an edge is able to predict the reduction in bias (or accuracy) upon pruning that edge. We do not make claims as to the goodness of the linear model specifically. A better way to substantiate this claim might be to consider whether the correlation between the two values is statistically significant:
>
> Dataset / Network | Flow type | Layer | Pearson correlation | p-value
> -----------------------|--------------|---------|--------------------------|-----------
> Synthetic / Small ANN | Accuracy | 1 |  $-0.35$ | $5.3 \times 10^{-27}$
>  |  | 2 |  $-0.20$ | $1.24 \times 10^{-6}$
>  | Bias | 1 |  $-0.47$ | $1.52 \times 10^{-50}$
>  |  | 2 |  $-0.03$ | $4.18 \times 10^{-1}$
> Modified Adult / Small ANN | Accuracy | 1 |  $-0.77$ | $1.81 \times 10^{-180}$
>  |  | 2 |  $-0.11$ | $6.89 \times 10^{-3}$
>  | Bias | 1 |  $-0.52$ | $1.37 \times 10^{-63}$
>  |  | 2 |  $-0.18$ | $5.21 \times 10^{-6}$
> Modified Adult / Large ANN | Accuracy | 1 |  $-0.57$ | $5.25 \times 10^{-132}$
>  |  | 2 |  $-0.51$ | $1.67 \times 10^{-61}$
>  |  | 3 |  $-0.04$ | $3.38 \times 10^{-1}$
>  | Bias | 1 |  $-0.62$ | $2.54 \times 10^{-160}$
>  |  | 2 |  $-0.48$ | $6.30 \times 10^{-52}$
>  |  | 3 |  $-0.10$ | $1.0 \times 10^{-2}$
>
> The table shows that the initial layer(s) show a large and statistically significant correlation between the information flow of every edge and the effect on the output accuracy/bias upon pruning that edge, substantiating our claim. We will include these results in the supplementary material as well.

---

> ### Author Response · Authors · 2021-08-12
> **Revised introductory paragraphs**
>
> Please find below the revised wording we intend to use in the final version for the introductory paragraphs. We have tried to take the reviewers' concerns into account, to streamline the example we use to motivate our paper, as well as to concretely connect the neuroscientific setting with the fairness context that is used as an analogy.
>
> > The "reward circuit" of the brain controls much of our behavior (Cooper et al., Neurotherapeutics 2017; Volkow et al., Physiological Reviews 2020). It is believed that addiction is characterized by a strong _bias_ towards immediate rewards in the reward circuit (Boettiger et al., J. Neurosci 2007). A different bias, called _affective bias_, in the same circuit, can lead to depression (Kilford et al., J. Affect. Disord. 2015). Broadly, such biases in the output of the reward circuit control our behavior and responses to stimuli. Recent technological advancements in neural probes and optogenetics (e.g., see Steinmetz et al., Science 2021; Boyden, F1000 Biol. Reports, 2011) are increasingly enabling us to record the reward circuit at a high resolution and _alter the network_ by activating or suppressing nodes and links, giving us powerful tools to understand and affect how this circuit processes information. Even a weak understanding of these tools is motivating clinical interventions for individuals suffering from depression, addiction, obesity, etc., and patients are starting to receive permanent electrode implants in brain regions that are involved in the reward circuit (Luigjes et al., Molecular Psychiatry, 2012; Delaloye et al., Dialogues Clin Neurosci. 2014; Border et al., Ment Illn. 2018; Formolo et al., Front. Neurosci. 2019). We identify a new and largely unexplored problem within this context: how do we perform _minimal interventions_ that "correct" undesirable biases without affecting other functions of this important network as much as possible?
> >
> > &nbsp;
> >
> > In this paper, we propose to use a recently developed framework for tracking the information flow about specific messages within a computational circuit (Venkatesh et al., IEEE Trans. Inf. Th. 2020) to identify interventions that can correct undesirable biases within the network. A key challenge that we face, however, is finding a sufficiently comprehensive dataset with simultaneous recordings of the various brain regions involved in the reward circuit. Although advances in experimental techniques suggest that we are on track to have such datasets in the coming years, simultaneous multi-area recordings of specific circuits are currently rare. To overcome this issue, we use the context of fairness in artificial intelligence to study the above question, which provides an excellent analogy to the neuroscientific context.
> >
> > &nbsp;
> >
> > The problem of introducing fairness in a decision-making system is much like that of removing undesirable biases from the reward circuit: (i) both systems rely on learned associations from stimuli (i.e., training data); (ii) both systems have an intended objective (to learn the association between the features and the true labels, or healthy reward-based learning); and (iii) both systems often learn undesirable associations (e.g., racial or gender bias in artificial neural networks, and biases that cause addiction, depression, or obsessive compulsive disorder in the reward circuit). This suggests that if we want to understand how information flows can inform interventions to correct biases in the reward network, we can simulate such experiments by examining the relationship between information flows and interventions in artificial neural networks trained on biased datasets.

---

### Author Response · Authors · 2021-08-19
**Information flow analysis on a deeper neural network with five hidden layers**

Dear reviewers,

Based on your comments, we also successfully replicated our results on a significantly deeper network with five hidden layers, trained on a part of the MNIST dataset. The network configuration was fully linear (with ReLU activation) had layer sizes [784, 6, 6, 6, 6, 6, 2]: 784 input features (flattened 28x28 MNIST images), 2 output neurons, and five hidden layers with 6 neurons each. For the dataset, we used four digits in two pairs: (4, 9) and (5, 8). These pairs were chosen based on a review of t-SNE plots of the MNIST dataset found in the literature: clusters corresponding to 4 and 9 were found to be very close to each other, as well as those corresponding to 5 and 8, suggesting the similarity between the two numbers in each pair. Since each pair also consists of one odd and one even number, we took the protected attribute Z to represent even vs. odd digits.

We skewed the training dataset to induce bias in the output, by having more training examples for 9 in the (4, 9) class, and more examples for 8 in the (5, 8) class at a 2:1 ratio. This maintained an equal number of examples of both classes while also having an equal number of odd and even examples overall. However, this skew induced the network to tend to prefer 9 over 4 at the output neuron for $\hat{Y} = 0$ and to prefer 8 over 5 for the output neuron for $\hat{Y} = 1$, allowing us to also guess whether the number was odd or even, based on $\hat{Y}$. The trained network was able to achieve approximately 98% accuracy and had a bias of around 65% (accuracy of classifying Z from $\hat{Y}$).

We performed the M-information flow analysis for all hidden layers and the output layer; we excluded the input layer for computational tractability. We also used correlation-based estimates for mutual information, as described in the supplementary material (see Appendix A.3 and Appendix D) for faster computation. Using our M-information flow measure to estimate the Y- and Z-information flows, we could once again produce tradeoff plots similar to those in Figures 3, 5 and 6. Based on the [Neurips FAQ](https://neurips.cc/Conferences/2021/PaperInformation/NeurIPS-FAQ), we are sharing these as an imgur link below, to preserve bidirectional anonymity.

[https://imgur.com/a/pgxrPuU](https://imgur.com/a/pgxrPuU)

The tradeoff plots closely mirror our expectations, based on similar results for the synthetic and modified Adult datasets. The dependency plots, however, show a very poor linear fit: but this is to be expected, considering that pruning individual edges are unlikely to have a significant impact on the output in such a large network.

We hope that the reviewers find these additional results convincing of the scalability of our approach to larger networks.

---

### Decision · Program_Chairs · 2021-09-27

**Decision:**

Accept (Poster)

**Comment:**

The paper considers how to use information flows to "debug" a neural net model toward desired fairness objectives. The idea seems novel and connects disparate fields in an interesting way. The reviewers appreciated the paper and agree it should be accepted, albeit their ratings were mostly borderline. The reviewers brought up many important concerns that the authors for the most part appropriately responded to. The authors are expected to address these points raised by reviewers in a final version, including as they outlined in their response and incorporate additional discussion or results from their responses into their paper in an appropriate manner.